# Adjoint Schrödinger Bridge Sampler

**Guan-Horng Liu**[1,*], **Jaemoo Choi**[2,*], **Yongxin Chen**[2], **Benjamin Kurt Miller**[1],
**Ricky T. Q. Chen**[1,*]

[1]FAIR at Meta, [2]Georgia Institute of Technology, *Core contributors

## Abstract

Computational methods for learning to sample from the Boltzmann distribution—
where the target distribution is known only up to an unnormalized energy function—
have advanced significantly recently. Due to the lack of explicit target samples,
however, prior diffusion-based methods, known as *diffusion samplers*, often require
importance-weighted estimation or complicated learning processes. Both trade off
scalability with extensive evaluations of the energy and model, thereby limiting their
practical usage. In this work, we propose **Adjoint Schrödinger Bridge Sampler
(ASBS)**, a new diffusion sampler that employs simple and scalable matching-
based objectives yet without the need to estimate target samples during training.
ASBS is grounded on a mathematical model—the Schrödinger Bridge—which
enhances sampling efficiency via kinetic-optimal transportation. Through a new
lens of stochastic optimal control theory, we demonstrate how SB-based diffusion
samplers can be learned at scale via Adjoint Matching and prove convergence to the
global solution. Notably, ASBS generalizes the recent Adjoint Sampling (Havens
et al., 2025) to arbitrary source distributions by relaxing the so-called memoryless
condition that largely restricts the design space. Through extensive experiments, we
demonstrate the effectiveness of ASBS on sampling from classical energy functions,
amortized conformer generation, and molecular Boltzmann distributions. Codes are
available at https://github.com/facebookresearch/adjoint_samplers.

## 1 Introduction

Sampling from Boltzmann distributions is a fundamental problem in computational science, with
widespread applications in Bayesian inference, statistical physics, and chemistry (Box and Tiao, 2011;
Binder et al., 1992; Tuckerman, 2023). Mathematically, we aim to sample from a target distribution
$\nu(x)$ known up to a unnormalized, often differentiable, energy function $E(x) : \mathcal{X} \subseteq \mathbb{R}^d \to \mathbb{R}$,

$$\nu(x) := \frac{e^{-E(x)}}{Z}, \qquad \text{where } \ Z := \int_{\mathcal{X}} e^{-E(x)} \mathrm{d}x \tag{1}$$

is an intractable normalization constant. For instance, the energy function $E(x)$ of a molecular system
quantifies the stability of a chemical structure based on the 3D positions of particles. A lower energy
indicates a more stable structure and hence a higher likelihood of its occurrence, *i.e.*, $\nu(x) \propto e^{-E(x)}$.

Classical methods that generate samples from $\nu(x)$ rely on Markov Chain Monte Carlo algorithms,
which run a Markov chain whose stationary distribution is $\nu(x)$ (Metropolis et al., 1953; Neal, 2001;
Del Moral et al., 2006). These methods, however, tend to suffer from slow mixing time and require
extensive evaluations of energy function, limiting their practical usages due to prohibitive complexity.

To improve sampling efficiency, modern samplers focus on learning better proposal distributions
(Noé et al., 2019; Midgley et al., 2023). Among those, recent advances in diffusion-based generative
models (Song et al., 2021; Ho et al., 2020) have given rise to a family of *Diffusion Samplers*, which

Table 1: Compared to prior diffusion samplers, **Adjoint Schrödinger Bridge Sampler** (**ASBS**) offers the most flexible design for diffusion samplers (2), while learning the drift $u_t^\theta$ via scalable matching objectives that do not rely on computation of importance weights (IWs).

| | Design condition for (2) | | Learning method for $u_t^\theta$ | |
| Method | Non-memoryless | Arbitrary prior | Matching objective[1] | No reliance on IWs |
|---|---|---|---|---|
| PIS (Zhang and Chen, 2022) DDS (Vargas et al., 2023) | ✗ | ✗ | ✗ | ✓ |
| LV-PIS & LV-DDS (Richter and Berner, 2024) | ✗ | ✗ | ✗ | ✗ |
| PDDS (Phillips et al., 2024) iDEM (Akhound-Sadegh et al., 2024) | ✗ | ✗ | ✓ | ✗ |
| AS (Havens et al., 2025) | ✗ | ✗ | ✓ | ✓ |
| Sequential SB (Bernton et al., 2019) | ✓ | ✓ | ✗ | ✗ |
| **Adjoint Schrödinger Bridge Sampler (Ours)** | ✓ | ✓ | ✓ | ✓ |

consider stochastic differential equations (SDEs) of the following form:

$$\mathrm{d}X_t = \left[f_t(X_t) + \sigma_t u_t^\theta(X_t)\right]\mathrm{d}t + \sigma_t \mathrm{d}W_t, \qquad X_0 \sim \mu(X_0), \tag{2}$$

where $f_t(x) : [0,1] \times \mathcal{X} \to \mathcal{X}$ the base drift, $\sigma_t : [0,1] \to \mathbb{R}_{>0}$ the noise schedule, and $\mu(x)$ the initial source distribution. Given $(f_t, \sigma_t, \mu)$, the diffusion sampler learns a parametrized drift $u_t^\theta(x)$ transporting samples to the target distribution $\nu(x)$ at the terminal time $t = 1$.

Computational methods for learning diffusion samplers have grown significantly recently (Zhang and Chen, 2022; Vargas et al., 2023; Berner et al., 2024; Chen et al., 2025). Due to the distinct problem setup in (1), the target distribution is defined exclusively by its energy $E(x)$, rather than by explicit target samples. This characteristic renders modern generative modeling techniques for scalability—particularly the score matching objectives[1]—less applicable. As such, prior matching-based diffusion samplers (Phillips et al., 2024; Akhound-Sadegh et al., 2024; De Bortoli et al., 2024) often require computationally intensive estimation of target samples via importance weights (IWs).

Recently, Havens et al. (2025) introduced Adjoint Sampling (AS), a new class of diffusion samplers whose matching objectives rely only on on-policy samples, thereby greatly enhancing scalability. By incorporating stochastic optimal control (SOC) theory (Kappen, 2005; Todorov, 2007), AS facilitates the use of Adjoint Matching (Domingo-Enrich et al., 2025), a novel matching objective that imposes self-consistency in generated samples, effectively eliminating the needs for target samples.

The efficiency of AS, however, is achieved through a specific instantiation of the SDE (2) to satisfy the so-called *memoryless* condition. This condition—formally discussed in Section 2—restricts its source distribution to be Dirac delta $\mu(x) := \delta$, precluding the use of common priors such as Gaussian or domain-specific priors such as the harmonic oscillators in molecular systems (Jing et al., 2023). Notably, the memoryless condition underlies *all* previous matching-based diffusion samplers, restricting the design space of (2) from other choices known to enhance transportation efficiency (Shaul et al., 2023). While the condition has been relaxed in non-matching-based methods at extensive computational complexity (Richter and Berner, 2024; Bernton et al., 2019), no existing diffusion sampler—to our best understanding—has successfully combined matching objectives with non-memoryless condition. Table 1 summarizes the comparison between prior diffusion samplers.

In this work, we propose **Adjoint Schrödinger Bridge Sampler (ASBS)**, a new adjoint-matching-based diffusion sampler that eliminates the requirement for memoryless condition entirely. Formally, ASBS recasts learning diffusion sampler as a distributionally constrained optimization, known as the Schrödinger Bridge (SB) problem (Schrödinger, 1931, 1932; Léonard, 2013; Chen et al., 2016):

$$\min_u D_{\mathrm{KL}}(p^u || p^{\mathrm{base}}) = \mathbb{E}_{X \sim p^u}\left[\int_0^1 \tfrac{1}{2}\|u_t^\theta(X_t)\|^2 \mathrm{d}t\right], \tag{3a}$$

$$\text{s.t. } \mathrm{d}X_t = \left[f_t(X_t) + \sigma_t u_t^\theta(X_t)\right]\mathrm{d}t + \sigma_t \mathrm{d}W_t, \qquad X_0 \sim \mu(X_0), \qquad X_1 \sim \nu(X_1). \tag{3b}$$

Here, $p^u$ denotes the path distribution induced by the SDE in (3b), whereas $p^{\mathrm{base}} := p^{u:=0}$ denotes the path distribution induced by the "base" SDE when $u_t := 0$. By minimizing their KL divergence, the SB problem (3) seeks the kinetic-optimal drift $u_t^\star$—an optimality structure well correlated

---

[1]The matching objective is a simple regression loss, $\mathbb{E}\|u_t^\theta(X_t) - v_t(X_t, X_1)\|^2$, w.r.t. some tractable $v_t$.

with sampling efficiency in generative modeling (Finlay et al., 2020; Liu et al., 2023). Since the SOC problem in AS corresponds to a specific case of the SB problem with $(f_t, \mu) := (0, \delta)$, ASBS extends AS to handle non-memoryless conditions by solving more general SB problems (see Theorem 3.1). Computationally, ASBS retains all scalability advantages from AS by utilizing an *adjoint*-matching objective that removes the need for estimating target samples. It also introduces a *corrector*-matching objective to correct nontrivial biases arising from non-memoryless conditions. We prove that alternating optimization between the two matching objectives is equivalent to executing the Iterative Proportional Fitting algorithm (Kullback, 1968), ensuring global convergence of ASBS to $u_t^\star$ (see Theorem 3.2). Though extensive experiments, we show superior performance of ASBS over prior diffusion samplers across various benchmarks on sampling multi-particle energy functions.

In summary, we present the following contributions:

- We introduce **ASBS**, an SB-based diffusion sampler capable of sampling target distributions using only unnormalized energy functions, by solving general SB problems with arbitrary priors.

- We base ASBS on a new SOC framework that removes the restrictive memoryless condition, develop a scalable matching-based algorithm, and prove theoretical convergence to global solution.

- We show ASBS's superior performance over prior methods on sampling Boltzmann distributions of classical energy functions, alanine dipeptide molecule and amortized conformer generation.

## 2 Preliminary

We revisit the memoryless condition introduced by Domingo-Enrich et al. (2025) and examine its impact on the constructions of SOC-based diffusion samplers (Zhang and Chen, 2022; Havens et al., 2025), which are closely related to our ASBS. Additional review can be found in Appendix A.

**Stochastic Optimal Control (SOC)** The SOC problem (4) studies an optimization problem:

$$\min_u \mathbb{E}_{X \sim p^u} \left[ \int_0^1 \tfrac{1}{2} \|u_t(X_t)\|^2 \mathrm{d}t + g(X_1) \right] \quad \text{s.t. (2),} \tag{4}$$

which, unlike the SB problem (3), includes an additional *terminal cost* $g(x) : \mathcal{X} \to \mathbb{R}$ at the terminal time $t = 1$ and considers the SDE without the terminal constraint $X_1 \sim \nu$. The primary reason for studying this specific optimization problem is that the optimal distribution is known analytically by[2]

$$p^\star(X_0, X_1) = p^{\text{base}}(X_0, X_1) e^{-g(X_1) + V_0(X_0)}, \quad \text{where} \quad V_0(x) = -\log \int p_{1|0}^{\text{base}}(y|x) e^{-g(y)} \mathrm{d}y \tag{5}$$

is the initial value function. That is, the optimal distribution $p^\star$ is an exponentially tilted version of the base distribution, $p^{\text{base}} := p^{u:=0}$. Specifically, $p^{\text{base}}$ is tilted by the terminal cost "$-g(X_1)$" and the initial value function $V_0(X_0)$, which is intractable. Consequently, to ensure its marginal $p^\star(X_1)$ follows the target distribution $\nu(X_1)$, we must eliminate the *initial value function bias* from $V_0(X_0)$.

**Memoryless condition & SOC-based diffusion sampler** A common approach to eliminate the aforementioned initial value function bias, adopted by most diffusion samplers, is to restrict the class of base processes to be *memoryless*. Formally, the memoryless condition assumes statistical independency between $X_0$ and $X_1$ in the base distribution:

$$p^{\text{base}}(X_0, X_1) \overset{\text{memoryless}}{:=} p^{\text{base}}(X_0) p^{\text{base}}(X_1). \tag{6}$$

This memoryless condition (6) simplifies the optimal distribution at the terminal time $t = 1$ and, upon choosing a proper terminal cost $g(x)$, recovers the target distribution $\nu$,

$$p^\star(X_1) \overset{\text{memoryless}}{=} \int p^{\text{base}}(X_0) p^{\text{base}}(X_1) e^{-g(X_1) + V_0(X_0)} \mathrm{d}X_0 \propto p^{\text{base}}(X_1) e^{-g(X_1)} = \nu(X_1),$$

where the last equality is due to setting the terminal cost to $g(x) := \log \frac{p_1^{\text{base}}(x)}{\nu(x)}$. Typically, the memoryless condition (6) is enforced by a careful design of the base distribution $p^{\text{base}}$ or, equivalently,

---

[2]Equation (5) can be obtained by rewriting (4) as $D_{\text{KL}}(p^u \| p^{\text{base}}) + \mathbb{E}_{p_1^u}[g(X_1)]$ and then computing the analytic solution $p^\star(X_1|X_0) \propto p^{\text{base}}(X_1|X_0) e^{-g(X_1)}$ and normalization $\int p^{\text{base}}(X_1|X_0) e^{-g(X_1)} \mathrm{d}X_1 = e^{-V_0(X_0)}$. See Appendix A.1 for details.

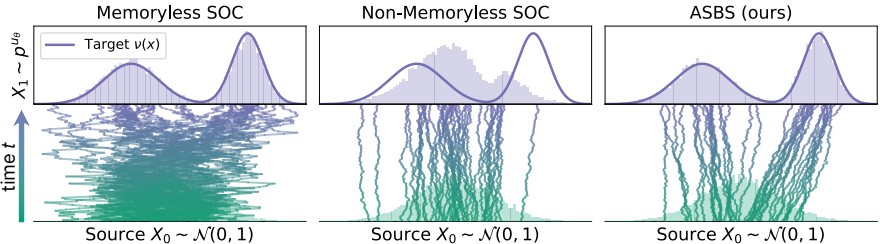

Figure 1: Effect of the memoryless condition on learning SOC-based diffusion samplers. We consider Gaussian prior $\mu(x) := \mathcal{N}(x; 0, 1)$ with $(f_t, \sigma_t)$ set to VP-SDE for the first plot and $(0, 0.2)$ for the rest; see Appendix A.1 for details. The memoryless condition injects significant noise (**left**) to correct the otherwise biased optimization (**middle**), whereas ASBS can successfully debias any non-memoryless processes (**right**).

the parameters $(f_t, \sigma_t, \mu)$ in (2). For instance, the variance-preserving process (VP; Song et al., 2021) considers a linear base drift $f_t$, a noise schedule $\sigma_t$ that grows significantly with time, and a Gaussian prior $\mu$; see Figure 1. Alternatively, one could implement (6) with Dirac delta prior $\mu(x) := \delta_0(x)$ and $f_t := 0$, leading to the following SOC problem (Zhang and Chen, 2022):

$$\min_{u} \mathbb{E}_{X \sim p^u} \left[ \int_0^1 \tfrac{1}{2} \|u_t(X_t)\|^2 \mathrm{d}t + \log \frac{p_1^{\text{base}}(X_1)}{\nu(X_1)} \right] \quad \text{s.t. } \mathrm{d}X_t = \sigma_t u_t(X_t)\mathrm{d}t + \sigma_t \mathrm{d}W_t, \quad X_0 = 0. \quad (7)$$

Based on the aforementioned reasoning, solving (7) results in a diffusion sampler that transports samples to the target distribution at $t=1$, with Adjoint Sampling (Havens et al., 2025) as the only scalable method of this class. Despite encouraging, the SOC problem in (7) is nevertheless limited by its trivial source, precluding potentially more effective options for sampling Boltzmann distributions.

## 3  Adjoint Schrödinger Bridge Sampler

We introduce a new diffusion sampler by solving the SB problem (3), where the target distribution $\nu(x)$ is given by its energy function $E(x)$ rather than explicit samples. All proofs are left in Appendix B.

### 3.1  SOC Characteristics of the SB Problem

The SB problem (3)—as an optimization problem with distribution constraints—is widely explored in optimal transport, stochastic control, and recently machine learning (Léonard, 2012; Chen et al., 2021; De Bortoli et al., 2021). Its kinetic-optimal drift $u^\star$ satisfies the following optimality equations:

$$u_t^\star(x) = \sigma_t \nabla \log \varphi_t(x), \quad \text{where} \quad \begin{cases} \varphi_t(x) = \int p_{1|t}^{\text{base}}(y|x)\varphi_1(y)\mathrm{d}y, & \varphi_0(x)\hat{\varphi}_0(x) = \mu(x) \quad (8a) \\[2mm] \hat{\varphi}_t(x) = \int p_{t|0}^{\text{base}}(x|y)\hat{\varphi}_0(y)\mathrm{d}y, & \varphi_1(x)\hat{\varphi}_1(x) = \nu(x) \quad (8b) \end{cases}$$

and $p_{t|s}^{\text{base}}(y|x) := p^{\text{base}}(X_t{=}y|X_s{=}x)$ is the transition kernel of the base process for observing $y$ at time $t$ given $x$ at time $s$. The *SB potentials* $\varphi_t(x), \hat{\varphi}_t(x) \in C^{1,2}([0, 1], \mathbb{R}^d)$ are then defined (up to some multiplicative constant) as solutions to forward and backward time integrations w.r.t. $p_{t|s}^{\text{base}}$.

Equation (8) are computationally challenging to solve—even when $p_{t|s}^{\text{base}}$ has an analytical solution—due to the intractable integration and coupled boundaries at $t = 0$ and $1$. Our key observation is that the first equation (8a) resembles the optimality condition of the SOC problem (4) (see Appendix A.1). This implies that the optimality conditions of SB hints an SOC reinterpretation, which, as we will demonstrate, is more tractable than solving (8) directly. We formalize our finding below.

**Theorem 3.1** (SOC characteristics of SB). *The kinetic-optimal drift $u_t^\star$ in (8) solves an SOC problem*

$$\min_{u} \mathbb{E}_{X \sim p^u} \left[ \int_0^1 \tfrac{1}{2} \|u_t(X_t)\|^2 \mathrm{d}t + \log \frac{\hat{\varphi}_1(X_1)}{\nu(X_1)} \right] \quad s.t. \ (2). \quad (9)$$

Theorem 3.1 suggests that *every* SB problem (3) can be solved like an SOC problem (4) with the terminal cost $g(x) := \log \frac{\hat{\varphi}_1(x)}{\nu(x)}$. Comparing to the formulation in Adjoint Sampling (Havens et al., 2025), the two SOC problems, namely (7) and (9), differ in their terminal costs—where $p_1^{\text{base}}$ is replaced by $\hat{\varphi}_1$—and the relaxation of the source distribution from Dirac delta $X_0 = 0$ to general source $\mu(X_0)$.

**How $\hat{\varphi}_1(\cdot)$ debiases non-memoryless SOC problems**    Taking a closer look at the effect of $\hat{\varphi}_1$, notice that the optimal distribution of the SB problem—according to Theorem 3.1 and (5)—follows

$$p^\star(X_0, X_1) = p^{\text{base}}(X_0, X_1) \exp\left(-\log \frac{\hat{\varphi}_1(X_1)}{\nu(X_1)} - \log \varphi_0(X_0)\right), \tag{10}$$

where "$-\log \varphi_0$" is the equivalent initial value function. One can verify that the marginal at the terminal time $t = 1$ indeed satisfies the target distribution,

$$
\begin{aligned}
p^\star(X_1) = \int p^\star(X_0, X_1) \mathrm{d}X_0 &\overset{(10)}{=} \frac{\nu(X_1)}{\hat{\varphi}_1(X_1)} \int p^{\text{base}}(X_0, X_1) \frac{1}{\varphi_0(X_0)} \mathrm{d}X_0 \\
&\overset{(8a)}{=} \frac{\nu(X_1)}{\hat{\varphi}_1(X_1)} \int p^{\text{base}}(X_1|X_0) \hat{\varphi}_0(X_0) \mathrm{d}X_0 \overset{(8b)}{=} \nu(X_1).
\end{aligned}
\tag{11}
$$

That is, the optimality equations in (8), in their essence, construct a specific function $\hat{\varphi}_1(\cdot)$ that eliminates the initial value function bias associated with any non-memoryless processes, thereby ensuring that the optimal distribution satisfies the target $\nu$ at $t = 1$.

## 3.2    Adjoint Sampling with General Source Distribution

We now specialize Theorem 3.1 to sampling Boltzmann distributions (1), where $\nu(x) \propto e^{-E(x)}$, and hence the terminal cost of the new SOC problem in (9) becomes $\log \frac{\hat{\varphi}_1(x)}{\nu(x)} = E(x) + \log \hat{\varphi}_1(x)$. To encourage minimal transportation cost (Chen and Georgiou, 2015; Peyré and Cuturi, 2017), we consider the Brownian-motion base process with a degenerate base drift $f_t := 0$. Applying Adjoint Matching (AM; Domingo-Enrich et al., 2025) to the resulting SOC problem leads to

$$u^\star = \arg\min_u \mathbb{E}_{p_{t|0,1}^{\text{base}} p_{0,1}^{\bar{u}}} \left[\|u_t(X_t) + \sigma_t \left(\nabla E + \nabla \log \hat{\varphi}_1\right)(X_1)\|^2\right], \quad \bar{u} = \texttt{stopgrad}(u). \tag{12}$$

Note that the AM objective in (12) functions as a self-consistency loss—in that both the regression and its expectation depend on the optimization variable $u$. This makes (12) particularly suitable for learning SB-based diffusion samplers, unlike previous matching-based SB methods (Shi et al., 2023; Liu et al., 2024), which all require ground-truth target samples from $X_1 \sim \nu$.

Computing the AM objective in (12) requires knowing $\nabla \log \hat{\varphi}_1(x)$, which, as we discussed in (11), serves as a *corrector* that debiases the optimization toward the desired target. Notably, this corrector function $\nabla \log \hat{\varphi}_1(x)$ also admits a variational form (Peluchetti, 2022, 2023; Shi et al., 2023):[3]

$$\nabla \log \hat{\varphi}_1 = \arg\min_h \mathbb{E}_{p_{0,1}^{u^\star}} \left[\|h(X_1) - \nabla_{x_1} \log p^{\text{base}}(X_1|X_0)\|^2\right]. \tag{13}$$

To summarize, Equations (12) and (13) characterize two distinct matching objectives that any kinetic-optimal drift $u_t^\star$ of SBs must satisfy. When the source distribution degenerates to Dirac delta $X_0 := 0$, (13) is minimized at $\nabla \log p_1^{\text{base}}$, and (12) simply recovers the objective used in Adjoint Sampling (Havens et al., 2025). In other words, (12) and (13) should be understood as a generalization of Adjoint Sampling to handle arbitrary—including *non-memoryless*—source distributions.

## 3.3    Alternating Optimization with Adjoint and Corrector Matching

Building upon the theoretical characterization in Section 3.2, we aim to design a learning algorithm that finds a diffusion sampler satisfying (12) and (13), which correspond to two simple matching-based objectives. However, these matching objectives cannot be naively implemented due to their interdependency: Solving (12) for the kinetic-optimal drift $u^\star$ requires knowing $\nabla \log \hat{\varphi}_1$. Likewise, solving (13) for the corrector function $\nabla \log \hat{\varphi}_1$ requires samples from $u^\star$. We relax the interdependency with an alternating optimization scheme. Specifically, given an approximation of $\nabla \log \hat{\varphi}_1 \approx h^{(k-1)}$ from the previous stage $k-1$, we first update the drift $u^{(k)}$ with the *AM* objective:

---

[3]Formally, $\nabla \log \hat{\varphi}_t(x)$ is the kinetic-optimal drift along the reversed time coordinate $s := 1 - t$, and (13) is its variational formulation, *i.e.,* the Markovian projection at $s = 0$; see Appendix A.2 for details.

---

**Algorithm 1** Adjoint Schrödinger Bridge Sampler (ASBS)

---

**Require:** Sample-able source $X_0 \sim \mu$, differentiable energy $E(x)$, parametrized $u_\theta(t,x)$ and $h_\phi(x)$

1: Initialize $h_\phi^{(0)} := 0$
2: **for** stage $k$ **in** $1, 2, \ldots$ **do**
3:     Update drift $u_\theta^{(k)}$ by solving (14)             ▷ adjoint matching
4:     Update corrector $h_\phi^{(k)}$ by solving (15)      ▷ corrector matching
5: **end for**

---

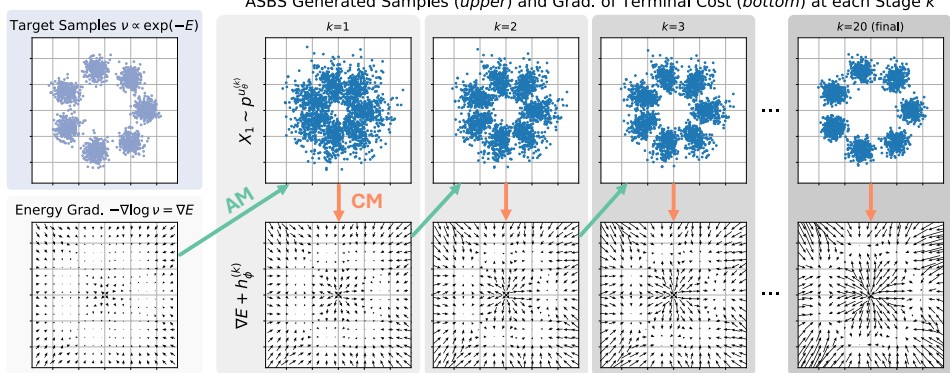

Figure 2: Illustration of ASBS on a 2D example. By alternatively minimizing the Adjoint Matching (AM) objective (14) and the Corrector Matching (CM) objective (15), ASBS progressively learns a better corrector $h_\phi^{(k)}$ that debiases the SOC problem for the control $u_\theta^{(k)}$. Note that since the corrector is initialized with $h_\phi^{(0)} := 0$, the first AM stage simply regresses $u_\theta^{(1)}$ to the energy gradient $\nabla E$.

$$u^{(k)} := \arg\min_u \mathbb{E}_{p_{t|0,1}^{\text{base}} p_{0,1}^{\bar{u}}} \left[ \|u_t(X_t) + \sigma_t(\nabla E + h^{(k-1)})(X_1)\|^2 \right], \quad \bar{u} = \texttt{stopgrad}(u). \quad (14)$$

Then, we use the resulting drift $u^{(k)}$ to update $h^{(k)}$ by minimizing the following matching objective, which—in light of the corrector role of $\nabla \log \hat{\varphi}_1$—we refer to as the *Corrector Matching* objective:

$$h^{(k)} := \arg\min_h \mathbb{E}_{p_{0,1}^{u^{(k)}}} \left[ \|h(X_1) - \nabla_{x_1} \log p^{\text{base}}(X_1|X_0)\|^2 \right]. \quad (15)$$

Equation (15) should be distinguish from the bridge-matching objectives in data-driven SB methods (Shi et al., 2023; Somnath et al., 2023), where $X_1$ must be drawn from the target distribution $\nu$. In contrast, the matching objectives in (14) and (15) depend only on model samples at the current stage $X_1 \sim p_\theta^{u^{(k)}}(X_1|X_0)$, hence can be used to learn SB-based diffusion samplers at scale.

The alternating optimization between (14) and (15) creates a sequence of updates, $(u^{(0)}, h^{(0)}) \to \cdots (u^{(k)}, h^{(k)}) \to \cdots$, that may be thought of as running coordinate descent between the control $u$ and the corrector $h$. Intuitively, at each stage $k$, we first find the control $u^{(k)}$ that best aligns with the corrector from previous stage, $h^{(k-1)}$, then update the corrector $h^{(k)}$ accordingly to reflect the "memorylessness" of the current control $u^{(k)}$. We summarize our method, **Adjoint Schrödinger Bridge Sampler (ASBS)**, in Algorithm 1, while leaving the full details with additional components, such as replay buffers, in Appendix C. Finally, we prove that this alternating optimization indeed converges to the kinetic-optimal drift $u^\star$ in (8).

**Theorem 3.2** (Global convergence of ASBS). *Algorithm 1 converges to the Schrödinger bridge solution of (3), provided all matching stages achieve their critical points, i.e.,*

$$\lim_{k \to \infty} u^{(k)} = u^\star.$$

# 4  Theoretical Analysis

We provide the proof of Theorem 3.2 and highlight theoretical insights throughout. While ASBS is specialized to a degenerate base drift $f_t := 0$, all theoretical results here apply to general $f_t$. To simplify notation, we omit the parameters $\theta$, $\phi$ and reparametrize the corrector by $h^{(k)} = \nabla \log \bar{h}^{(k)}$. All proofs are left in Appendix B.

Our first result presents a variational characteristic to the solution of the AM objective in (14).

**Theorem 4.1** (Adjoint Matching solves a forward half bridge). *Let $p^{u^{(k)}}$ be the path distribution induced by the drift $u^{(k)}$ in (14) at stage $k$. Then, $p^{u^{(k)}}$ solves the following variational problem:*

$$p^{u^{(k)}} = \arg\min_p \left\{ D_{\mathrm{KL}}(p || q^{\bar{h}^{(k-1)}}) : p_0 = \mu \right\}, \tag{16}$$

*where $q^{\bar{h}^{(k-1)}}$ is the path distribution induced by a "backward" SDE on the reversed time coordinate $s := 1 - t$, defined by the corrector from the previous stage $\bar{h}^{(k-1)}$:*

$$\mathrm{d}Y_s = \left[ -f_s(Y_s) + \sigma_s^2 \nabla \log \phi_s(Y_s) \right] \mathrm{d}s + \sigma_s \mathrm{d}W_s, \quad \phi_s(y) = \int p_{1-s|0}^{\mathrm{base}}(y|z) \phi_1(z) \mathrm{d}z, \tag{17}$$

*with the boundary conditions $Y_0 \sim \nu$ and $\phi_0(y) = \bar{h}^{(k-1)}(y)$.*

Theorem 4.1 suggests that any SOC problems with the terminal cost $g(x) := \log \frac{\bar{h}^{(k)}(x)}{\nu(x)}$ can be reinterpreted as KL minimization w.r.t. a specific *backward* SDE (17) that is fully characterized by $\nu$—which serves as its source distribution—and $\bar{h}^{(k)}$—which defines its drift through the function $\phi_s(y)$. The objective in (16) differs from the one in the original SB problem (3) by disregarding the target boundary constraint, $X_1 \sim \nu$. Consequently, (16) only solves a forward half bridge.

Next, we show that the CM objective (15) admits a similar variational form, except backward in time.

**Theorem 4.2** (Corrector Matching solves a backward half bridge). *Let $\bar{h}^{(k)}$ be the corrector in (15) at stage $k$. Then, the path distribution $q^{\bar{h}^{(k)}}$ solves the following variational problem:*

$$q^{\bar{h}^{(k)}} = \arg\min_q \left\{ D_{\mathrm{KL}}(p^{u^{(k)}} || q) : q_1 = \nu \right\} \tag{18}$$

Unlike (16), the objective in (18) disregards the source boundary constraint $\mu$ instead, thereby solving a backward half bridge. Theorems 4.1 and 4.2 imply that our ASBS in Algorithm 1 *implicitly* employs an optimization scheme that alternates between solving forward and backward half bridges, thereby instantiating the celebrated Iterative Proportional Fitting algorithm (IPF; Fortet, 1940; Kullback, 1968). Combining with the analysis by (De Bortoli et al., 2021) leads to our final result in Theorem 3.2.

# 5  Related Works

We provide additional clarification on SB-related works and leave the full review to Appendix A.3.

**Data-driven Schrödinger Bridges**  The SB problem has attracted notable interests in machine learning due to its connection to diffusion-based generative models (Wang et al., 2021). Earlier methods implemented classical IPF algorithms (De Bortoli et al., 2021; Vargas et al., 2021; Chen et al., 2022), with scalability later enhanced by bridge matching-based methods (Shi et al., 2023; Liu et al., 2024). Unlike ASBS, all of them focus on generative modeling and assume access to extensive target samples during training, making them unsuitable for sampling from Boltzmann distributions.

**SB-inspired Diffusion Samplers**  Notably, in the context of diffusion samplers, the SB formulation has been constantly emphasized as a mathematically appealing framework for both theoretical analysis and method motivation (Zhang and Chen, 2022; Vargas et al., 2024; Richter and Berner, 2024; Havens et al., 2025). None of the prior methods, however, offers general solutions to learning SB-based diffusion samplers, instead specializing to either the memoryless condition or non-matching-based objectives, which largely complicate the learning process (see Table 1). Conceptually, our ASBS stands closest to SSB (Bernton et al., 2019) by learning general SB samplers. However, the two methods differ fundamentally in scalability: SSB is a Sequential Monte Carlo-based method (Chopin, 2002) augmented with learned transition kernels using Gaussian-approximated SB potentials. As with many MCMC-augmented samplers (Gabrié et al., 2022; Matthews et al., 2022), SSB requires extensive evaluations on the energy $E(x)$, in contrast to ASBS, which is much more energy-efficient.

Table 2: Results on the synthetic energy functions for $n$-particle bodies with their corresponding dimensions $d$. Following (Chen et al., 2025; Havens et al., 2025), we report Sinkhorn for MW-5 and the Wasserstein-2 distances w.r.t samples, $\mathcal{W}_2$, and energies, $E(\cdot)\mathcal{W}_2$, for the rest. All values are averaged over three random trials. Best results are highlighted.

| Method | MW-5 ($d$=5) Sinkhorn $\downarrow$ | DW-4 ($d = 8$) $\mathcal{W}_2 \downarrow$ | $E(\cdot)\,\mathcal{W}_2 \downarrow$ | LJ-13 ($d = 39$) $\mathcal{W}_2 \downarrow$ | $E(\cdot)\,\mathcal{W}_2 \downarrow$ | LJ-55 ($d = 165$) $\mathcal{W}_2 \downarrow$ | $E(\cdot)\,\mathcal{W}_2 \downarrow$ |
|---|---|---|---|---|---|---|---|
| PDDS (Phillips et al., 2024) | — | $0.92_{\pm0.08}$ | $0.58_{\pm0.25}$ | $4.66_{\pm0.87}$ | $56.01_{\pm10.80}$ | — | — |
| SCLD (Chen et al., 2025) | $0.44_{\pm0.06}$ | $1.30_{\pm0.64}$ | $0.40_{\pm0.19}$ | $2.93_{\pm0.19}$ | $27.98_{\pm\,1.26}$ | — | — |
| PIS (Zhang and Chen, 2022) | $0.65_{\pm0.25}$ | $0.68_{\pm0.28}$ | $0.65_{\pm0.25}$ | $1.93_{\pm0.07}$ | $18.02_{\pm\,1.12}$ | $4.79_{\pm0.45}$ | $228.70_{\pm131.27}$ |
| DDS (Vargas et al., 2023) | $0.63_{\pm0.24}$ | $0.92_{\pm0.11}$ | $0.90_{\pm0.37}$ | $1.99_{\pm0.13}$ | $24.61_{\pm\,8.99}$ | $4.60_{\pm0.09}$ | $173.09_{\pm\,18.01}$ |
| LV-PIS (Richter and Berner, 2024) | — | $1.04_{\pm0.29}$ | $1.89_{\pm0.89}$ | — | — | — | — |
| iDEM (Akhound-Sadegh et al., 2024) | — | $0.70_{\pm0.06}$ | $0.55_{\pm0.14}$ | $1.61_{\pm0.01}$ | $30.78_{\pm24.46}$ | $4.69_{\pm1.52}$ | $93.53_{\pm\,16.31}$ |
| AS (Havens et al., 2025) | $0.32_{\pm0.06}$ | $0.62_{\pm0.06}$ | $0.55_{\pm0.12}$ | $1.67_{\pm0.01}$ | $2.40_{\pm\,1.25}$ | $4.04_{\pm0.05}$ | $30.83_{\pm\,8.19}$ |
| ASBS (**Ours**) | $0.15_{\pm0.02}$ | $0.43_{\pm0.05}$ | $0.20_{\pm0.11}$ | $1.59_{\pm0.03}$ | $1.99_{\pm\,1.01}$ | $4.00_{\pm0.03}$ | $28.10_{\pm\,8.15}$ |

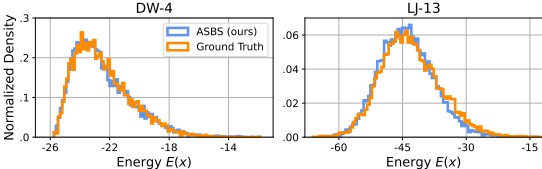
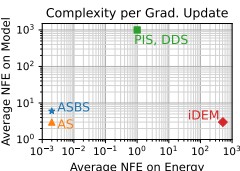

Figure 3: The energy histograms of DW-4 and LJ-13 from Table 2. ASBS generates samples whose energy profiles closely match those of the ground-truth samples.

Figure 4: Complexity w.r.t. the number of function evaluation (NFE) on LJ-13 potential.

## 6 Experiments

**Benchmarks**   We evaluate our ASBS on three classes of multi-particle energy functions $E(x)$.

- *Synthetic energy functions*   These are classical potentials based on pair-wise distances of an $n$-particle system, where $E(x)$ is known analytically. Following (Akhound-Sadegh et al., 2024; Chen et al., 2025), we consider a 2D 4-particle Double-Well potential (DW-4), a 1D 5-particle Many-Well potential (MW-5), a 3D 13-particle Lennard-Jones potential (LJ-13) and a 3D 55-particle Lennard-Jones potential (LJ-55). For the ground-truth samples, we sample analytically from MW-5 and use the MCMC samples from (Klein et al., 2023) for the rest of three potentials.

- *Alanine dipeptide*   This is a molecule consisting of 22 atoms in 3D. Specifically, we consider the alanine dipeptide in an implicit solvent and aim to sample from its Boltzmann distribution at a temperature $300K$. Following prior methods (Zhang and Chen, 2022; Wu et al., 2020), we use the energy function $E(x)$ from the OpenMM library (Eastman et al., 2017) and consider a more structural internal coordinate with the dimension $d = 60$. The ground-truth samples contain $10^7$ configurations, simulated from Molecular Dynamics (Midgley et al., 2023).

- *Amortized conformer generation*   Finally, we consider a new benchmark proposed in (Havens et al., 2025) for large-scale conformer generation. Conformers are locally stable configurations located at the local minima of the molecule's potential energy surface (Hawkins, 2017). Sampling conformers is essentially a conditional generation task, targeting a Boltzmann distribution $\nu(x|g) \propto e^{-\frac{1}{\tau}E(x|g)}$ at a low temperature $\tau \ll 1$, conditioned on the molecular topology $g \in \mathcal{G}$. The training set $\mathcal{G}_{\text{train}}$ contains 24,477 molecular topologies from SPICE (Eastman et al., 2023), represented by the SMILES strings (Weininger, 1988), whereas the test set $\mathcal{G}_{\text{test}}$ contains 80 topologies from SPICE and another 80 from GEOM-DRUGS (Axelrod and Gomez-Bombarelli, 2022). As with (Havens et al., 2025), we consider $E(x|g)$ a foundation model *eSEN* from (Fu et al., 2025), which predicts energy with density-functional-theory accuracy at a much lower computational cost. We use CREST conformers (Pracht et al., 2024) as the ground-truth samples.

**Baselines and evaluation**   We compare ASBS with a wide range of diffusion samplers, including PIS (Zhang and Chen, 2022), DDS (Vargas et al., 2023), PDDS (Phillips et al., 2024), SCLD (Chen

Table 3: Comparison between diffusion samplers on sampling the molecular Boltzmann distribution of the alanine dipeptide. We report the KL divergence $D_{\text{KL}}$ for the 1D marginal across five torsion angles and the Wasserstein-2 $\mathcal{W}_2$ on jointly $(\phi, \psi)$, known as Ramachandran plots (see Figure 5). Best results are highlighted.

| | $D_{\text{KL}}$ on each torsion's marginal ↓ | | | | | $\mathcal{W}_2$ on joint ↓ |
|---|---|---|---|---|---|---|
| Method | $\phi$ | $\psi$ | $\gamma_1$ | $\gamma_2$ | $\gamma_3$ | $(\phi, \psi)$ |
| PIS (Zhang and Chen, 2022) | $0.05_{\pm 0.03}$ | $0.38_{\pm 0.49}$ | $5.61_{\pm 1.24}$ | $4.49_{\pm 0.03}$ | $4.60_{\pm 0.03}$ | $1.27_{\pm 1.19}$ |
| DDS (Vargas et al., 2023) | $0.03_{\pm 0.01}$ | $0.16_{\pm 0.07}$ | $2.44_{\pm 0.96}$ | $0.03_{\pm 0.00}$ | $0.03_{\pm 0.00}$ | $0.68_{\pm 0.09}$ |
| AS (Havens et al., 2025) | $0.09_{\pm 0.09}$ | $0.04_{\pm 0.04}$ | $0.17_{\pm 0.17}$ | $0.56_{\pm 0.09}$ | $0.51_{\pm 0.06}$ | $0.65_{\pm 0.52}$ |
| ASBS (**Ours**) | $0.02_{\pm 0.00}$ | $0.01_{\pm 0.00}$ | $0.03_{\pm 0.01}$ | $0.02_{\pm 0.00}$ | $0.02_{\pm 0.00}$ | $0.25_{\pm 0.01}$ |

Table 4: Results on large-scale amortized conformer generation, evaluated on two test sets, SPICE and GEOM-DRUGS, both with and without post-processing relaxation. We report the coverage (%) and Absolute Mean RMSD (AMR) of the recall at the threshold **1.0Å**. Note that "*+RDKit warmup*" refers to warm-starting the model $u_\theta$ using RDKit conformers; see Appendix D for details. Best results without and with RDKit warm-up are highlighted separately.

| | without relaxation | | | | with relaxation | | | |
|---|---|---|---|---|---|---|---|---|
| | SPICE | | GEOM-DRUGS | | SPICE | | GEOM-DRUGS | |
| Method | Coverage ↑ | AMR ↓ | Coverage ↑ | AMR ↓ | Coverage ↑ | AMR ↓ | Coverage ↑ | AMR ↓ |
| RDKit ETKDG (Riniker and Landrum, 2015) | $56.94_{\pm 35.82}$ | $1.04_{\pm 0.52}$ | $50.81_{\pm 34.69}$ | $1.15_{\pm 0.61}$ | $70.21_{\pm 31.70}$ | $0.79_{\pm 0.44}$ | $62.55_{\pm 31.67}$ | $0.93_{\pm 0.53}$ |
| AS (Havens et al., 2025) | $56.75_{\pm 38.15}$ | $0.96_{\pm 0.26}$ | $36.23_{\pm 33.42}$ | $1.20_{\pm 0.43}$ | $82.41_{\pm 25.85}$ | $0.68_{\pm 0.28}$ | $64.26_{\pm 34.57}$ | $0.89_{\pm 0.45}$ |
| ASBS w/ Gaussian prior (**Ours**) | $73.04_{\pm 31.95}$ | $0.83_{\pm 0.24}$ | $50.23_{\pm 35.98}$ | $1.05_{\pm 0.43}$ | $88.26_{\pm 20.57}$ | $0.60_{\pm 0.24}$ | $72.32_{\pm 29.68}$ | $0.77_{\pm 0.35}$ |
| ASBS w/ harmonic prior (**Ours**) | $74.05_{\pm 31.61}$ | $0.82_{\pm 0.23}$ | $53.14_{\pm 35.69}$ | $1.03_{\pm 0.42}$ | $88.71_{\pm 18.63}$ | $0.59_{\pm 0.24}$ | $72.77_{\pm 29.94}$ | $0.78_{\pm 0.35}$ |
| AS +RDKit warmup (Havens et al., 2025) | $72.21_{\pm 30.22}$ | $0.84_{\pm 0.24}$ | $52.19_{\pm 35.20}$ | $1.02_{\pm 0.34}$ | $87.84_{\pm 19.20}$ | $0.60_{\pm 0.23}$ | $73.88_{\pm 28.63}$ | $0.76_{\pm 0.34}$ |
| ASBS +RDKit warmup (**Ours**) | $77.84_{\pm 28.37}$ | $0.79_{\pm 0.23}$ | $57.19_{\pm 35.14}$ | $0.98_{\pm 0.40}$ | $88.08_{\pm 18.84}$ | $0.58_{\pm 0.24}$ | $73.18_{\pm 30.09}$ | $0.76_{\pm 0.37}$ |

et al., 2025), LV (Richter and Berner, 2024), iDEM (Akhound-Sadegh et al., 2024) and finally Adjoint Sampling (AS; Havens et al., 2025). For the conformer generation task, we include additionally a domain-specific baseline, RDKit ETKDG (Riniker and Landrum, 2015), which relies on chemistry-based heuristics. The evaluation pipelines are consistent with prior methods, where we adopt the SCLD setup for MW-5, the PIS setup for alanine dipeptide, and the AS setup for all the rest; see Appendix D for details.

**ASBS models** For all tasks, we consider a degenerate base drift $f_t := 0$, as discussed in Section 3.2, and set $\sigma_t$ a geometric noise schedule. For energy functions that directly take particle systems as inputs—such as DW, LJ, and eSEN—we parametrize the models $u_\theta$, $h_\phi$ with two Equivariant Graph Neural Networks (Satorras et al., 2021) and consider a domain-specific source distribution—the harmonic prior (Jing et al., 2023). Formally, for an $n$-particle system $x = \{x_i\}_{i=0}^n$, the harmonic prior $\mu_{\text{harmonic}}(x)$ is a quadratic potential that can be sampled analytically from an anisotropic Gaussian:

$$\mu_{\text{harmonic}}(x) \propto \exp\left(-\tfrac{\alpha}{2} \sum_{i,j} \|x_i - x_j\|^2\right). \tag{19}$$

For other energy functions, we use standard fully-connected neural networks and consider Gaussian priors. All models are trained with Adam (Kingma and Ba, 2015) and, following standard practices (Havens et al., 2025; Akhound-Sadegh et al., 2024), utilize replay buffers; see Appendix C for details.

**Results** Table 2 presents the results on synthetic energy functions. Notably, ASBS consistently outperforms prior diffusion samplers across *all* energy functions. In Figure 3, we compare the energy histograms of DW-4 and LJ-13 potentials between the ground-truth MCMC samples and those from ASBS. It is evident that ASBS generates samples that closely resemble the target Boltzmann distribution $\nu(x) \propto e^{-E(x)}$, resulting in energy profiles $E(x)$ that are almost indistinguishable from the ground truth. Computationally, Figure 4 shows the average number of evaluation required on the energy $E(x)$ and the model $u_\theta(t, x)$ for each gradient update. ASBS is much more efficient than most diffusion samplers, with a slight overhead compared to AS due to the additional network $h_\phi(x)$.

Table 3 summarizes the results for alanine dipeptide. Following standard pipeline (Zhang and Chen, 2022), we generate model samples $X_1 \in \mathbb{R}^{60}$ and extract five torsion angles—including the backbone

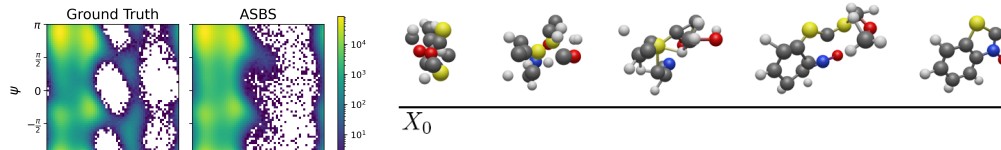

Figure 5: Ramachandran plots for the alanine dipeptide between ground-truth and ASBS samples.

Figure 6: Example of ASBS generative process on amortized conformer generation. Given an unseen molecular topology $g \in \mathcal{G}_{\text{test}}$ from the test set—COCSc1sc2ccccc2[n+]1[O-] in this case—ASBS transports samples from the harmonic prior $X_0 \sim \mu_{\text{harmonic}}$ to generate conformers $X_1$.

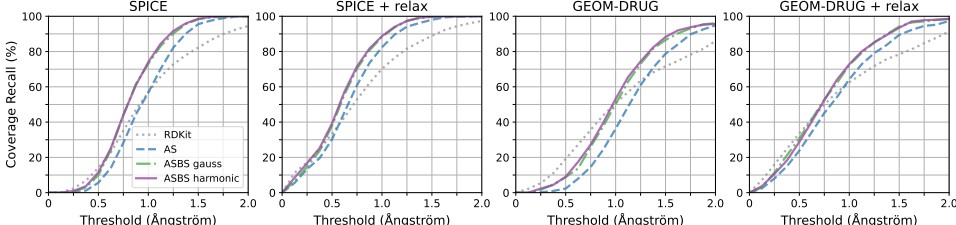

Figure 7: Recall coverage curves on amortized conformer generation on the SPICE and GEOM-DRUGS test sets without RDKit warm-start. Note that Table 4 reports the recall coverages at the threshold **1.0Å**.

angles $\phi$, $\psi$ and methyl rotation angles $\gamma_1$, $\gamma_2$, $\gamma_3$—all of them exhibit multi-modal distributions. Notably, ASBS achieves lowest KL divergence to the ground-truth marginals across all five torsions. Figure 5 further compares the joint distributions of $(\phi, \psi)$, known as the Ramachandran plots (Spencer et al., 2019), between ground-truth and ASBS. While ASBS identifies all high-density modes in the region $\phi \in [-\pi, 0]$, it misses few low-density modes. This mode-seeking behavior, inherit in all SOC-based diffusion samplers, could be improved with important weighting. We provide further discussions in Appendix D.4.

Table 4 presents the recall for amortized conformer generation compared to ground-truth samples. For prior diffusion samplers, we primarily compare to AS (Havens et al., 2025) due to the benchmark's scale. Following AS, we ablate a warm-start stage using RDKit conformers, which are close but not identical to ground-truth samples, and include results with relaxation for post-generation optimization. Since AS is a specific instance of ASBS with a Dirac delta prior—as discussed in Section 3.2—any performance improvements from AS to ASBS highlight the added capability to handle arbitrary priors and, consequently, non-memoryless processes. Remarkably, without any warm-start, ASBS with the harmonic prior (19) already matches and, in many cases, surpasses the RDKit-warm-up AS. With warm-start, ASBS achieves best performance across most metrics. This highlights the significance of domain-specific priors, aiding exploration as effectively as warm-start with additional data, which may not always be available. Finally, we visualize the generation process of ASBS with harmonic prior (19) in Figure 6 and report the recall curves in Figure 7. In practice, we observe that ASBS achieves slightly better results with a harmonic prior compared to a Gaussian prior, with both significantly outperforming AS (Havens et al., 2025). See Appendix D.4 for further ablation studies.

# 7  Conclusion and Limitation

We introduced **Adjoint Schrödinger Bridge Sampler (ASBS)**, a new diffusion sampler for Boltzmann distributions that solves general SB problems given only target energy functions. ASBS is based on a scalable matching framework, converges theoretically to the global solution, and performs superiorly across various benchmarks. Despite these encouraging results, further enhancement with importance sampling techniques is worth investigating to mitigate the mode collapse inherent in SOC-inspired diffusion samplers. Exploring its effectiveness in sampling amortized Boltzmann distributions would also be valuable.

## Acknowledgements

The authors would like to thank Aaron Havens, Juno Nam, Xiang Fu, Bing Yan, Brandon Amos, and Brian Karrer for the helpful discussions and comments. JC and YC acknowledge support from NSF Grants ECCS-1942523, DMS-2206576, and CMMI-2450378.

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

# A  Additional Preliminary and Reviews

## A.1  Stochastic Optimal Control (SOC)

In this subsection, we expand Section 2 with details. Recall the SOC problem in (4):

$$\min_{u} \mathbb{E}_{X\sim p^u}\left[\int \tfrac{1}{2}\|u_t(X_t)\|^2 \mathrm{d}t + g(X_1)\right] \tag{20a}$$

$$\text{s.t. } \mathrm{d}X_t = [f_t(X_t) + \sigma_t u_t(X_t)]\,\mathrm{d}t + \sigma_t \mathrm{d}W_t, \;\; X_0 \sim \mu. \tag{20b}$$

Similar to (8), the optimal control to (20) can be characterized through an optimality equation:

$$u_t^\star(x) = -\sigma_t \nabla V_t(x), \quad \text{where} \quad V_t(x) = -\log \int p_{1|t}^{\text{base}}(y|x)e^{-V_1(y)}\mathrm{d}y, \quad V_1(x) = g(x) \tag{21}$$

is the value function known to satisfy the Hamilton–Jacobi–Bellman (HJB) equation (Bellman, 1954). We provide further characterization below.

**Optimal distribution**  The optimization problem in (20) is known analytically. Specifically, notice that the entropy-regularized objective in (20) can be reformulated as:

$$D_{\text{KL}}(p(X)\|p^{\text{base}}(X)) + \mathbb{E}_{p(X)}\left[g(X_1)\right]$$

$$= D_{\text{KL}}\left(p(X_0)\|p^{\text{base}}(X_0)\right) + \mathbb{E}_{p(X_0)}\left[D_{\text{KL}}\left(p(X|X_0)\|p^{\text{base}}(X|X_0)\right) + \mathbb{E}_{p(X|X_0)}\left[g(X_1)\right]\right]$$

$$= D_{\text{KL}}\left(p(X_0)\|p^{\text{base}}(X_0)\right) + \mathbb{E}_{p(X_0)}\left[D_{\text{KL}}\left(p(X|X_0)\|p^{\text{base}}(X|X_0)e^{-g(X_1)}\right)\right] \tag{22}$$

where we shorthand $X \equiv X_{[0,1]}$ and denote $p^{\text{base}}$ the base distribution induced by (20b) with $u := 0$, *i.e.*, the uncontrolled distribution. Minimizing (22) w.r.t. $p$ yields

$$p^\star(X|X_0) = \frac{1}{Z(X_0)}p^{\text{base}}(X|X_0)e^{-g(X_1)}, \qquad p^\star(X_0) = p^{\text{base}}(X_0) \tag{23}$$

where $Z(X_0)$ is the normalization term defined by

$$Z(X_0) := \int p^{\text{base}}(X|X_0)e^{-g(X_1)}\mathrm{d}X = \int p^{\text{base}}(X_1|X_0)e^{-g(X_1)}\mathrm{d}X_1 \tag{24}$$

which is exactly $e^{-V(X_0)}$ due to (21). Combing (23) and (24) leads to the the optimal distribution in (5), which we restate below for completeness:

$$p^\star(X) = p^{\text{base}}(X)e^{-g(X_1)+V_0(X_0)} \implies p^\star(X_0, X_1) = p^{\text{base}}(X_0, X_1)e^{-g(X_1)+V_0(X_0)} \tag{25}$$

**Adjoint Matching (AM)**  Scalable computational methods for solving (20) have been challenging, as naively back-propagating through (20) induces prohibitively high computational cost. Instead, Adjoint Matching (Domingo-Enrich et al., 2025) employs a matching-based objective, named Adjoint Matching (AM):

$$u^\star = \arg\min_{u} \mathbb{E}_{X\sim p^{\bar{u}}}\left[\|u_t(X_t) + \sigma_t a_t\|^2\right], \qquad \bar{u} = \texttt{stopgrad}(u), \tag{26a}$$

$$\text{where} \quad -\mathrm{d}a_t = a_t \cdot \nabla f_t(X_t)\mathrm{d}t, \quad a_1 = \nabla g(X_1) \tag{26b}$$

is the backward dynamics of the (lean) adjoint state $a_t \equiv a(t; X_{[t,1]})$. It has been proven that the unique critical point of (26) is the optimal control $u^\star$, implying a new characteristics of the optimal control $u^\star$ using the adjoint state:

$$u_t^\star(x) = -\sigma_t \mathbb{E}_{p^\star}[a_t|X_t = x]. \tag{27}$$

**Adjoint Sampling (AS)**  Recently, Havens et al. (2025) introduced an adaptation of AM tailored to sampling Boltzmann distribution $\nu(x) \propto e^{-E(x)}$ by considering

$$f_t := 0, \qquad \mu(x) := \delta_0(x), \qquad g(x) := \log \frac{p_1^{\text{base}}(x)}{\nu(x)}. \tag{28}$$

That is, AS considers the following SOC problem with a degenerate base drift, a Dirac delta prior, and a specific instantiation of the terminal cost $g(x) := \log \frac{p_1^{\text{base}}(x)}{\nu(x)}$:

$$\min_u \mathbb{E}_{X \sim p^u} \left[ \int_0^1 \frac{1}{2} \|u_t(X_t)\|^2 \mathrm{d}t + \log \frac{p_1^{\text{base}}(X_1)}{\nu(X_1)} \right] \quad \text{s.t. } \mathrm{d}X_t = \sigma_t u_t(X_t) \mathrm{d}t + \sigma_t \mathrm{d}W_t, \ X_0 = 0. \quad (29)$$

Notably, this SOC problem (29) admits a simplified adjoint state $a_t$ and a degenerate initial value function $V_0(x)$:

$$a_t \overset{(26b)}{=} \nabla g(X_1) \overset{(28)}{=} \nabla \log p_1^{\text{base}}(X_1) + \nabla E(X_1) \qquad \forall t \in [0, 1] \quad (30)$$

$$V_0(x) \overset{(28)}{=} -\log \int p_1^{\text{base}}(y) \frac{\nu(y)}{p_1^{\text{base}}(y)} \mathrm{d}y = -\log 1 = 0, \quad (31)$$

which further implies that the optimal distribution $p^\star$ is a reciprocal process (Léonard et al., 2014):

$$p^\star(X) \overset{(31)}{=} p^{\text{base}}(X) e^{-V_1(X_1)} \overset{(28)}{=} p^{\text{base}}(X) \frac{\nu(X_1)}{p_1^{\text{base}}(X_1)} = p^{\text{base}}(X|X_1) p^\star(X_1). \quad (32)$$

Combining the adjoint characteristics of the optimal control (27) with the simplified adjoint state $a_t$ in (30) and optimal distribution $p^\star$ (32) motivates the following *Reciprocal Adjoint Matching (RAM)* objective used in AS, where the unique critical point remains to be the optimal control $u^\star$ in (21).

$$u^\star = \arg\min_u \mathbb{E}_{p_{t|1}^{\text{base}} p_1^{\bar{u}}} \left[ \|u_t(X_t) + \sigma_t \left( \nabla E + \nabla \log p_1^{\text{base}} \right) (X_1)\|^2 \right], \quad \bar{u} = \texttt{stopgrad}(u). \quad (33)$$

**Remark on reciprocal representation** The reciprocal representation of the optimal-controlled distribution $p^\star$ in (32) extends to general SOC problems (20) with non-trivial base drifts and source distributions. Specifically, any optimal-controlled distribution that solves (20) can be factorized by

$$p^\star(X) = p^{\text{base}}(X|X_0, X_1) p^\star(X_0, X_1). \quad (34)$$

We leave a formal statement in Theorem B.3 and Corollary B.4.

**AS with linear base drift and Gaussian prior (Figure 1)** Here, we discuss an alternative instantiation of AM for sampling with linear base drift and Gaussian prior, which reproduces the leftmost plot in Figure 1. Consider

$$f_t(x) := -\frac{1}{2} \beta_t x, \qquad \mu(x) := \mathcal{N}(x; 0, I), \qquad \sigma_t := \sqrt{\beta_t}, \qquad g(x) := \log \frac{p_1^{\text{base}}(x)}{\nu(x)}. \quad (35)$$

where $\beta_t$ is chosen such that $(f_t, \mu, \sigma_t)$ fulfill the memoryless condition. For instance, Figure 1 adopts the VPSDE (Song et al., 2021) setup:

$$\beta_t = (1-t)\beta_{\max} + t\beta_{\min}, \qquad \beta_{\max} = 20, \qquad \beta_{\min} = 0.1. \quad (36)$$

Similar to (30), the resulting SOC problem admits a simplified adjoint state $a_t$:

$$a_t \overset{(26b)}{=} \kappa_t \cdot \nabla g(X_1) \overset{(35)}{=} \kappa_t \cdot (\nabla \log p_1^{\text{base}}(X_1) + \nabla E(X_1)), \qquad \kappa_t := e^{-\frac{1}{2} \int_t^1 \beta_\tau \mathrm{d}\tau} \overset{(36)}{=} e^{-\frac{1}{4}(1-t)(\beta_t + \beta_1)} \quad (37)$$

and the RAM objective becomes

$$u^\star = \arg\min_u \mathbb{E}_{p_{t|0,1}^{\text{base}} p_{0,1}^{\bar{u}}} \left[ \|u_t(X_t) + \sigma_t \kappa_t \left( \nabla E + \nabla \log p_1^{\text{base}} \right) (X_1)\|^2 \right], \quad \bar{u} = \texttt{stopgrad}(u). \quad (38)$$

Note that $p_{t|0,1}^{\text{base}}$ can be sampled analytically:

$$p_{t|0,1}^{\text{base}}(X_t|X_0, X_1) \overset{(35)}{=} \mathcal{N}(X_t; \frac{\bar{\kappa}_t(1 - \kappa_t^2)}{1 - \bar{\kappa}_1^2} X_0 + \frac{\kappa_t(1 - \bar{\kappa}_t^2)}{1 - \bar{\kappa}_1^2} X_1, \frac{(1 - \kappa_t^2)(1 - \bar{\kappa}_t^2)}{1 - \bar{\kappa}_1^2} I), \quad (39)$$

where $\kappa_t$ is defined in (37) and $\bar{\kappa}_t := e^{-\frac{1}{2} \int_0^t \beta_\tau \mathrm{d}\tau} \overset{(36)}{=} e^{-\frac{1}{4}t(\beta_t + \beta_0)}$.

## A.2 Schrödinger Bridge (SB)

In this subsection, we provide additional clarification on SB and specifically the derivation of (13). Recall the optimality equations of SB in (8):

$$u_t^\star(x) = \sigma_t \nabla \log \varphi_t(x), \quad \text{where} \begin{cases} \varphi_t(x) = \int p_{1|t}^{\text{base}}(y|x)\varphi_1(y)\mathrm{d}y, & \varphi_0(x)\hat{\varphi}_0(x) = \mu(x) \quad \text{(40a)} \\[2mm] \hat{\varphi}_t(x) = \int p_{t|0}^{\text{base}}(x|y)\hat{\varphi}_0(y)\mathrm{d}y, & \varphi_1(x)\hat{\varphi}_1(x) = \nu(x) \quad \text{(40b)} \end{cases}$$

Just like how the value function of an SOC problem fully characterizes the optimal control and its corresponding optimal distribution, so does the SB potential $\varphi_t(x)$:

$$p^\star(X) = p^{\text{base}}(X)\frac{\varphi_1(X_1)}{\varphi_0(X_0)} = p^{\text{base}}(X|X_0)\varphi_1(X_1)\hat{\varphi}_0(X_0), \tag{41}$$

where the last equality is due to $p^{\text{base}}(X) = p^{\text{base}}(X|X_0)\mu(X_0)$ and then invoking (40a). Note that (41) recovers (10) by marginalizing over $t \in (0,1)$. Due to the construction of $\varphi_t(x)$ and $\hat{\varphi}_t(x)$ in (40), the marginal optimal distribution admits a strikingly simple factorization:

$$\begin{aligned} p_t^\star(x) &= \int p^{\text{base}}(X, X_t = x|X_0)\varphi_1(X_1)\hat{\varphi}_0(X_0)\mathrm{d}X \\ &= \int\int p^{\text{base}}(X_1|X_t = x)p^{\text{base}}(X_t = x|X_0)\varphi_1(X_1)\hat{\varphi}_0(X_0)\mathrm{d}X_0\mathrm{d}X_1 \\ &= \left(\int p^{\text{base}}(X_t = x|X_0)\hat{\varphi}_0(X_0)\mathrm{d}X_0\right)\left(\int p^{\text{base}}(X_1|X_t = x)\varphi_1(X_1)\mathrm{d}X_1\right) \\ &= \hat{\varphi}_t(x)\varphi_t(x), \end{aligned} \tag{42}$$

or, more generally,

$$p_{s,t}^\star(y, x) = p_{t|s}^{\text{base}}(x|y)\hat{\varphi}_s(y)\varphi_t(x), \qquad s \leq t. \tag{43}$$

**Derivation of** (13)    We now provide a simpler derivation of (13) compared to its original derivation based on path measure theory (Shi et al., 2023):

$$\begin{aligned} \nabla \log \hat{\varphi}_t(x) &\overset{(40b)}{=} \frac{1}{\hat{\varphi}_t(x)}\nabla_x\int p_{t|0}^{\text{base}}(x|y)\hat{\varphi}_0(y)\mathrm{d}y \\ &= \frac{1}{\hat{\varphi}_t(x)}\int \nabla_x \log p_{t|0}^{\text{base}}(x|y)p_{t|0}^{\text{base}}(x|y)\hat{\varphi}_0(y)\mathrm{d}y \\ &= \int \nabla_x \log p_{t|0}^{\text{base}}(x|y)p_{0|t}^\star(y|x)\mathrm{d}y, \end{aligned} \tag{44}$$

where the last equality follows by

$$p_{0|t}^\star(y|x) \overset{(42)}{=} \frac{p_{0,t}^\star(y, x)}{\hat{\varphi}_t(x)\varphi_t(x)} \overset{(43)}{=} \frac{p_{t|0}^{\text{base}}(x|y)\hat{\varphi}_0(y)\varphi_t(x)}{\hat{\varphi}_t(x)\varphi_t(x)} = \frac{p_{t|0}^{\text{base}}(x|y)\hat{\varphi}_0(y)}{\hat{\varphi}_t(x)}.$$

Equation (44) implies a matching-based variational formulation of $\nabla \log \hat{\varphi}_t(\cdot)$—also known as the *bridge matching* objective in data-driven SB (Shi et al., 2023; Liu et al., 2023).

$$\nabla \log \hat{\varphi}_t = \underset{h}{\arg\min} \, \mathbb{E}_{p_{0,t}^\star}\left[\|h_t(X_t) - \nabla_{x_t} \log p^{\text{base}}(X_t|X_0)\|^2\right]. \tag{45}$$

Equation (45) recovers (13) at $t = 1$.

## A.3 Additional Related Works

In this subsection, we provide additional review on existing learning-based methods for sampling Boltzmann distributions.

**Learning-augmented MCMC**    This class of methods can be thought of as extension of classical sampling methods—such as MCMC (Metropolis et al., 1953; HASTINGS, 1970), Sequential Monte Carlo (SMC; Del Moral et al., 2006) and Annealed Importance Sampling (AIS; Neal, 2001)—where

traditional proposal distributions are replaced with modern machine learning models. For instance, Arbel et al. (2021) and Gabrié et al. (2022) use normalizing flows (Chen et al., 2018) as learned proposal distributions, whereas Matthews et al. (2022) employ stochastic normalizing flow (Wu et al., 2020). More recently, Chen et al. (2025) have explored the use of diffusion models (Song et al., 2021; Ho et al., 2020). However, training these models typically requires computing importance weights, which necessitates a large number of energy evaluations.

**MCMC-augmented Diffusion Samplers** Alternatively, methods of this class adopt modern generative models to sampling Boltzmann distributions and incorporate MCMC techniques to mitigate the lack of explicit target samples. For example, Phillips et al. (2024), (De Bortoli et al., 2024) and (Akhound-Sadegh et al., 2024) employ score matching objective from score-based diffusion models (Song et al., 2021; Ho et al., 2020). In contrast, Albergo and Vanden-Eijnden (2024) base their method on action matching objectives (Neklyudov et al., 2023). However, estimating target samples requires computing importance weights, which makes these methods computationally expensive in terms of energy function evaluations.

# B Proofs

## B.1 Preliminary and Additional Theoretical Results

**Lemma B.1** (Itô lemma (Itô, 1951)). *Let $X_t$ be the solution to the Itô SDE:*

$$\mathrm{d}X_t = f_t(X_t)\mathrm{d}t + \sigma_t \mathrm{d}W_t.$$

*Then, the stochastic process $v_t(X_t)$, where $v \in C^{1,2}([0,1],\mathbb{R}^d)$, is also an Itô process:*

$$\mathrm{d}v_t(X_t) = \left[\partial_t v_t(X_t) + \nabla v_t(X_t) \cdot f + \frac{1}{2}\sigma_t^2 \Delta v_t(X_t)\right]\mathrm{d}t + \sigma_t \nabla v_t(X_t) \cdot \mathrm{d}W_t. \tag{46}$$

**Lemma B.2** (Laplacian trick). *For any twice-differentiable function $\pi$ such that $\pi(x) \neq 0$, it holds that*

$$\frac{1}{\pi(x)}\Delta\pi(x) = \|\nabla \log \pi(x)\|^2 + \Delta \log \pi(x) \tag{47}$$

*Proof.*

$$\begin{aligned}
\Delta\pi(x) &= \nabla \cdot \nabla\pi(x) \\
&= \nabla \cdot (\pi(x)\nabla \log \pi(x)) \\
&= \nabla\pi(x) \cdot \nabla \log \pi(x) + \pi(x)\Delta \log \pi(x) \\
&= \pi(x)\left(\|\nabla \log \pi(x)\|^2 + \Delta \log \pi(x)\right)
\end{aligned}$$

$\square$

**Theorem B.3** (SB characteristics of SOC). *The optimal distribution $p^\star$ of the SOC problem in (20) is also the solution to the following SB problem:*

$$\arg\min_p \left\{ D_{\mathrm{KL}}(p\|p^{\mathrm{base}}) : p_0 = \mu, \quad p_1 = p_1^\star \right\}. \tag{48}$$

*Proof.* We aim to show that there exist a transform such that the SOC's optimality equation (21) can be reinterpreted as the ones for SB (40). To this end, consider

$$\varphi_t(x) := e^{-V_t(x)}, \qquad \hat\varphi_t(x) := e^{V_t(x)}p_t^\star(x). \tag{49}$$

One can verify that the value function $V_t(x)$ defined in (21) can be rewritten as

$$\varphi_t(x) = \int p_{1|t}^{\mathrm{base}}(y|x)\varphi_1(y)\mathrm{d}y.$$

On the other hand, we can expand $\hat{\varphi}_t(x)$ by

$$
\begin{aligned}
\hat{\varphi}_t(x) &= e^{V_t(x)} \int p^\star(X|X_t = x)\mathrm{d}X \\
&= e^{V_t(x)} \int p^{\text{base}}(X_1|X_t = x)p^{\text{base}}(X_t = x, X_0)e^{-V_1(X_1)+V_0(X_0)}\mathrm{d}X_1\mathrm{d}X_0 && \text{by (25)} \\
&= e^{V_t(x)} \int p^{\text{base}}(X_t = x, X_0)e^{-V_t(x)+V_0(X_0)}\mathrm{d}X_0 && \text{by (21)} \\
&= \int p^{\text{base}}(X_t = x|X_0)\mu(X_0)e^{V_0(X_0)}\mathrm{d}X_0 \\
&= \int p_{t|0}^{\text{base}}(x|y)\hat{\varphi}_0(y)\mathrm{d}y. && \text{by (49)}
\end{aligned}
$$

Combined, the optimality equation (21) for the SOC problem can be rewritten equivalently as

$$
u_t^\star(x) = \sigma_t \nabla \log \varphi_t(x), \quad \text{where} \quad
\begin{cases}
\varphi_t(x) = \int p_{1|t}^{\text{base}}(y|x)\varphi_1(y)\mathrm{d}y, & \varphi_0(x)\hat{\varphi}_0(x) = \mu(x), \\
\hat{\varphi}_t(x) = \int p_{t|0}^{\text{base}}(x|y)\hat{\varphi}_0(y)\mathrm{d}y, & \varphi_1(x)\hat{\varphi}_1(x) = p_1^\star(x).
\end{cases}
$$

We conclude that $p^\star$ indeed solves (48). $\qquad\square$

**Corollary B.4** (Reciprocal process of the SOC problem)**.** *The optimal distribution $p^\star$ of the SOC problem in* (20) *is a reciprocal process, i.e.,*

$$
p^\star(X) = p^{\text{base}}(X|X_0, X_1)p^\star(X_0, X_1). \tag{51}
$$

## B.2  Missing Proofs in Main Paper

**Proof of Theorem 3.1**  Comparing (8a) to (21), we can reinterpret $\varphi_t(x)$ as an value function $V_t(x)$ by reinterpreting

$$
V_t(x) := -\log \varphi_t(x), \quad g(x) := -\log \varphi_1(x) \overset{(8b)}{=} \log \frac{\hat{\varphi}_1(x)}{\nu(x)}.
$$

That is, the kinetic-optimal drift of SB solves an SOC problem (4) with a terminal cost $g(x) := \frac{\hat{\varphi}_1(x)}{\nu(x)}$. $\square$

**Proof of Theorem 4.1**  For notational simplicity, we will denote $q \equiv q^{\bar{h}^{(k-1)}}$ throughout the proof. We first rewrite the backward SDE (17) in the forward direction (Nelson, 2020):

$$
\mathrm{d}X_t = \left[f_t - \sigma_t^2 \nabla \log \phi_t + \sigma_t^2 \nabla \log q_t\right]\mathrm{d}t + \sigma_t \mathrm{d}W_t, \quad X_1 \sim \nu,
$$

where we rewrite $\phi_t(x)$ w.r.t. the forward time coordiante:

$$
\phi_t(x) = \int p_{t|0}^{\text{base}}(x|y)\phi_0(y)\mathrm{d}y, \qquad \phi_1(x) = \bar{h}^{(k-1)}(x). \tag{52}
$$

Note that (52) admits an equivalent PDE form by invoking Feynman-Kac formula (Le Gall, 2016):

$$
\partial_t \phi_t(x) = -\nabla \cdot (f_t \phi_t) + \frac{\sigma_t^2}{2}\Delta \phi_t(x), \quad \phi_1(x) = \bar{h}^{(k-1)}(x). \tag{53}
$$

On the other hand, the dynamics of $\partial_t q$ follows the Fokker Plank equation (Øksendal, 2003):

$$
\begin{aligned}
\partial_t q_t &= -\nabla \cdot \left(\left(f_t - \sigma_t^2 \nabla \log \phi_t + \sigma_t^2 \nabla \log q_t\right)q_t\right) + \frac{1}{2}\sigma_t^2 \Delta q_t \\
&= \nabla \cdot \left(\left(\sigma_t^2 \nabla \log \phi_t - f_t\right)q_t\right) - \frac{1}{2}\sigma_t^2 \Delta q_t,
\end{aligned}
$$

and straightforward calculation yields

$$
\partial_t \log q_t = \sigma_t^2 \Delta \log \phi_t - \nabla \cdot f_t + \left(\sigma_t^2 \nabla \log \phi_t - f_t\right) \cdot \nabla \log q_t - \frac{1}{2}\sigma_t^2 \|\nabla \log q_t\|^2 - \frac{1}{2}\sigma_t^2 \Delta \log q_t,
$$
$$
\tag{54}
$$

where we apply the Laplacian trick (47) to $\frac{1}{q}\Delta q = \|\nabla \log q_t\|^2 + \Delta \log q_t$.

Now, recall that $p$ is the path distribution induced by the following SDE:

$$\mathrm{d}X_t = [f_t(X_t) + \sigma_t u_t(X_t)]\,\mathrm{d}t + \sigma_t \mathrm{d}W_t, \qquad X_0 \sim \mu. \tag{55}$$

Invoke Ito Lemma (46) to $\log q_t(X_t)$, where $X_t$ follows (55):

$$\mathrm{d}\log q_t = \left[\partial_t \log q_t + \nabla \log q_t \cdot (f_t + \sigma_t u_t) + \frac{1}{2}\sigma_t^2 \Delta \log q_t\right]\mathrm{d}t + \sigma_t \nabla \log q_t \cdot \mathrm{d}W_t$$

$$\overset{(54)}{=} \left[\sigma_t^2 \Delta \log \phi_t - \nabla \cdot f_t + \sigma_t^2 \nabla \log \phi_t \cdot \nabla \log q_t - \frac{1}{2}\sigma_t^2\|\nabla \log q_t\|^2 + \nabla \log q_t \cdot (\sigma_t u_t)\right]\mathrm{d}t$$

$$+ \sigma_t \nabla \log q_t \cdot \mathrm{d}W_t \tag{56}$$

Likewise, invoke Ito Lemma (46) to $\log \phi_t(X_t)$, where $X_t$ follows (55):

$$\mathrm{d}\log \phi_t$$

$$= \left[\partial_t \log \phi_t + \nabla \log \phi_t \cdot (f_t + \sigma_t u_t) + \frac{1}{2}\sigma_t^2 \Delta \log \phi_t\right]\mathrm{d}t + \sigma_t \nabla \log \phi_t \cdot \mathrm{d}W_t$$

$$\overset{(53)}{=} \left[-\nabla \cdot f_t + \frac{\sigma_t^2}{2}\frac{\Delta \phi_t}{\phi_t} + \nabla \log \phi_t \cdot (\sigma_t u_t) + \frac{1}{2}\sigma_t^2 \Delta \log \phi_t\right]\mathrm{d}t + \sigma_t \nabla \log \phi_t \cdot \mathrm{d}W_t$$

$$\overset{(47)}{=} \left[-\nabla \cdot f_t + \frac{\sigma_t^2}{2}\left(\|\nabla \log \phi_t\|^2 + \Delta \log \phi_t\right) + \nabla \log \phi_t \cdot (\sigma_t u_t) + \frac{1}{2}\sigma_t^2 \Delta \log \phi_t\right]\mathrm{d}t + \sigma_t \nabla \log \phi_t \cdot \mathrm{d}W_t$$

$$= \left[-\nabla \cdot f_t + \frac{\sigma_t^2}{2}\|\nabla \log \phi_t\|^2 + \nabla \log \phi_t \cdot (\sigma_t u_t) + \sigma_t^2 \Delta \log \phi_t\right]\mathrm{d}t + \sigma_t \nabla \log \phi_t \cdot \mathrm{d}W_t \tag{57}$$

Subtracting (57) from (56) leads to

$$\mathrm{d}\log \phi_t - \mathrm{d}\log q_t = \left[\frac{1}{2}\|u_t + \sigma_t \nabla \log \phi_t - \sigma_t \nabla \log q_t\|^2 - \frac{1}{2}\|u_t\|^2\right]\mathrm{d}t + \sigma_t \nabla \log \frac{\phi_t}{q_t} \cdot \mathrm{d}W_t. \tag{58}$$

Finally, we are ready to compute the variational objective in (16):

$$D_{\mathrm{KL}}(p\|q^{\bar{h}^{(k-1)}}) = \mathbb{E}_{X \sim p^u}\left[\int_0^1 \frac{1}{2}\|u_t(X_t) + \sigma_t \nabla \log \phi_t(X_t) - \sigma_t \nabla \log q_t(X_t)\|^2 \mathrm{d}t\right]$$

$$\overset{(58)}{=} \mathbb{E}_{X \sim p^u}\left[\int_0^1 \left(\frac{1}{2}\|u_t(X_t)\|^2 + \mathrm{d}\log \phi_t(X_t) - \mathrm{d}\log q_t(X_t)\right)\mathrm{d}t\right]$$

$$= \mathbb{E}_{X \sim p^u}\left[\int_0^1 \frac{1}{2}\|u_t(X_t)\|^2 \mathrm{d}t + \log \frac{\phi_1(X_1)}{q_1(X_1)} - \log \frac{\phi_0(X_0)}{q_0(X_0)}\right] \tag{59}$$

$$\propto \mathbb{E}_{X \sim p^u}\left[\int_0^1 \frac{1}{2}\|u_t(X_t)\|^2 \mathrm{d}t + \log \frac{\bar{h}^{(k-1)}(X_1)}{\nu(X_1)}\right]. \tag{60}$$

That is, we have shown that the variational objective $D_{\mathrm{KL}}(p\|q^{\bar{h}^{(k-1)}})$ is equivalent (up to an additive constant) to an SOC problem (60). Applying Reciprocal Adjoint Matching (Havens et al., 2025) with the reciprocal process from Corollary B.4 conclude that $D_{\mathrm{KL}}(p\|q^{\bar{h}^{(k-1)}})$ is minimized by $p^{u^{(k)}}$. $\square$

**Proof of Theorem 4.2** For notational simplicity, we will denote $p^{(k)} \equiv p^{u^{(k)}}$ throughout the proof. Let $q$ be the path distribution induced by a backward SDE, propagating along the time coordinate $s := 1 - t$:

$$\mathrm{d}Y_s = [-f_s(Y_s) + \sigma_s v_s(Y_s)]\,\mathrm{d}s + \sigma_s \mathrm{d}W_s, \qquad Y_0 \sim \nu.$$

Next, rewrite the forward SDE $p^{(k)}$ in the backward direction:

$$\mathrm{d}Y_s = \left[-f_s - \sigma_s u_s^{(k)} + \sigma_s^2 \nabla \log p_s^{(k)}\right]\mathrm{d}s + \sigma_s \mathrm{d}W_s, \quad Y_0 \sim p_t^{(k)}|_{t=1}.$$

By Theorem B.3, we know that $p^{(k)}$ is the SB solution, thereby satisfying

$$u_t^{(k)}(x) = \sigma_t \nabla \log \varphi_t(x), \text{ where } \begin{cases} \varphi_t(x) = \int p_{1|t}^{\mathrm{base}}(y|x)\varphi_1(y)\mathrm{d}y, \ \varphi_0(x)\hat{\varphi}_0(x) = \mu(x) & \text{(61a)} \\[2mm] \hat{\varphi}_t(x) = \int p_{t|0}^{\mathrm{base}}(x|y)\hat{\varphi}_0(y)\mathrm{d}y, \ \varphi_1(x)\hat{\varphi}_1(x) = p_1^{(k)}(x) & \text{(61b)} \end{cases}$$

Since we are working with the backward time coordinate $s$, it is convenience to define $\phi_s := \hat{\varphi}_{1-t}$ and rewrite (61b) by

$$\phi_s(y) = \int p_{1-s|0}^{\text{base}}(y|z)\phi_1(z)\mathrm{d}z, \quad \phi_0(y) = \frac{p_1^{(k)}(y)}{\varphi_1(y)}. \tag{62}$$

Now, expanding the variational objective with Girsanov Theorem yields (Särkkä and Solin, 2019)

$$D_{\text{KL}}(p^{(k)}||q) = \mathbb{E}_{Y \sim p^{(k)}} \left[ \int_0^1 \tfrac{1}{2}\| - \sigma_s\nabla\log\varphi_s(Y_s) + \sigma_s\nabla\log p_s^{(k)}(Y_s) - v_s(Y_s)\|^2\mathrm{d}s \right], \tag{63}$$

which is minimized point-wise at

$$v_s^\star(y) = \sigma_s\nabla\log\frac{p_s^{(k)}(y)}{\varphi_s(y)} \overset{(42)}{=} \sigma_s\nabla\log\hat{\varphi}_s(y).$$

In other words, the backward SDE that minimizes (63) must obey

$$\mathrm{d}Y_s = \left[ -f_s(Y_s) + \sigma_s^2\nabla\log\phi_s(Y_s) \right]\mathrm{d}s + \sigma_s\mathrm{d}W_s, \quad Y_0 \sim \nu,$$

with $\phi_s$ defined in (62). That is, we have concluded so far that

$$q^{p_1^{(k)}/\varphi_1} = \arg\min_q \left\{ D_{\text{KL}}(p^{(k)}||q) : q_1 = \nu \right\}. \tag{64}$$

Hence, it remains to be shown that the minimizer $\bar{h}_1^{(k)}$ of the CM objective at stage $k$ equals $\frac{p_1^{(k)}}{\varphi_1}$. This is indeed the case since $p^{(k)}$ is the SB solution:

$$\nabla\log\bar{h}^{(k)} \overset{(15)}{:=} \arg\min_h \mathbb{E}_{p_{0,1}^{(k)}} \left[ \|h(X_1) - \nabla_{x_1}\log p^{\text{base}}(X_1|X_0)\|^2 \right] \overset{(45)}{=} \nabla\log\hat{\varphi}_1 \overset{(42)}{=} \nabla\log\frac{p_1^{(k)}}{\varphi_1}.$$

$$\square$$

## C  Practical Implementation of ASBS

Algorithm 2 summarizes the practical implementation of ASBS, where we expand the adjoint and corrector matching steps (*i.e.,* lines 3 and 4 in Algorithm 1) to full details. Table 5 provides the hyper-parameters for each task. We break down each component as follows:

**Harmonic prior** $\mu_{\text{harmonic}}$    Recall the harmonic prior in (19):

$$\mu_{\text{harmonic}}(x) \propto \exp(-\tfrac{1}{2}\sum_{i,j}\|x_i - x_j\|^2). \tag{65}$$

In practice, we set $\alpha = 1$ and implement (65) as an anisotropic Gaussian. For instance, for a 2-particle system in 3D, *i.e.,* $x = [x_1; x_2] \in \mathbb{R}^6$, we can rewrite (65) as a quadratic potential,

$$\exp(-\tfrac{1}{2}\|x_1 - x_2\|^2) = \exp(x^\top R x), \quad \text{where } R = \begin{bmatrix} 1 & 0 & 0 & -\tfrac{1}{2} & 0 & 0 \\ 0 & 1 & 0 & 0 & -\tfrac{1}{2} & 0 \\ 0 & 0 & 1 & 0 & 0 & -\tfrac{1}{2} \\ -\tfrac{1}{2} & 0 & 0 & 1 & 0 & 0 \\ 0 & -\tfrac{1}{2} & 0 & 0 & 1 & 0 \\ 0 & 0 & -\tfrac{1}{2} & 0 & 0 & 1 \end{bmatrix}, \tag{66}$$

and then sample $x$ from the Gaussian $\mathcal{N}(x; 0, (R + \epsilon I)^{-1})$, where we set $\epsilon = 10^{-4}$.

**Noise schedule** $\sigma_t$    We consider two types of noise schedule.

- The *geometric noise schedule* (Song et al., 2021; Karras et al., 2022) monotonically decays from $t = 0$ to 1 according to some prescribed $\beta_{\text{min}}$ and $\beta_{\text{max}}$:

$$\sigma_t \overset{\text{geometric}}{:=} \beta_{\text{min}}\left(\frac{\beta_{\text{max}}}{\beta_{\text{min}}}\right)^{1-t}\sqrt{2\log\frac{\beta_{\text{max}}}{\beta_{\text{min}}}}. \tag{67}$$

---

**Algorithm 2** Adjoint Schrödinger Bridge Sampler (ASBS)

---

**Require:** Sample-able source $X_0 \sim \mu$, differentiable energy $E(x)$, parametrized drift $u_\theta(t, x)$ and corrector $h_\phi(x)$, replay buffers $\mathcal{B}_{\text{adj}}$ and $\mathcal{B}_{\text{crt}}$, number of stages $K$, numbers of AM and CM epochs $M_{\text{adj}}$ and $M_{\text{crt}}$, number of resamples $N$, number of gradient steps $L$, time scaling $\lambda_t$, maximum energy gradient norm $\alpha_{\max}$.

1: Initialize $h_\phi^{(0)} := 0$                    ▷ IPF initialization
2: **for** stage $k$ **in** $1, 2, \ldots, K$ **do**
3:     **for** epoch **in** $1, 2, \ldots, M_{\text{adj}}$ **do**               ▷ adjoint matching
4:         Sample from model $\{(X_0^{(i)}, X_1^{(i)})\}_{i=1}^N \sim p^{\bar{u}^{(k)}}$, where $\bar{u}^{(k)} = \texttt{stopgrad}(u_\theta^{(k)})$
5:         Compute adjoint target $a_t^{(i)} := \texttt{stopgrad}\left(\texttt{clip}(\nabla E(X_1^{(i)}), \alpha_{\max}) + h_\phi^{(k)}(X_1^{(i)})\right)$
6:         Update replay buffer $\mathcal{B}_{\text{adj}} \leftarrow \mathcal{B}_{\text{adj}} \cup \{(X_0^{(i)}, X_1^{(i)}, a_t^{(i)})\}_{i=1}^N$
7:         Take $L$ gradient steps $\nabla_\theta \mathcal{L}_{\text{AM}}$ w.r.t. the AM objective:

$$\mathcal{L}_{\text{AM}}(\theta) := \mathbb{E}_{t \sim \mathcal{U}[0,1], (X_0, X_1, a_t) \sim \mathcal{B}_{\text{adj}}, X_t \sim p^{\text{base}}(\cdot | X_0, X_1)} \left[ \lambda_t \| u_\theta^{(k)}(t, X_t) + \sigma_t a_t \|^2 \right]$$

8:     **end for**

9:     **for** epoch **in** $1, 2, \ldots, M_{\text{crt}}$ **do**               ▷ corrector matching
10:        Sample from model $\{(X_0^{(i)}, X_1^{(i)})\}_{i=1}^N \sim p^{\bar{u}^{(k)}}$, where $\bar{u}^{(k)} = \texttt{stopgrad}(u_\theta^{(k)})$
11:        Update replay buffer $\mathcal{B}_{\text{crt}} \leftarrow \mathcal{B}_{\text{crt}} \cup \{(X_0^{(i)}, X_1^{(i)})\}_{i=1}^N$
12:        Take $L$ gradient steps $\nabla_\phi \mathcal{L}_{\text{CM}}$ w.r.t. the CM objective:

$$\mathcal{L}_{\text{CM}}(\phi) := \mathbb{E}_{(X_0, X_1) \sim \mathcal{B}_{\text{crt}}} \left[ \| h_\phi^{(k)}(X_1) - \nabla_{x_1} \log p^{\text{base}}(X_1 | X_0) \|^2 \right]$$

13:     **end for**
14: **end for**
15: **return** Kinetic-optimal drift $u^\star \approx u_\theta(t, x)$

---

It is convenience to further define

$$\kappa_{t|s} := \int_s^t \sigma_\tau^2 d\tau \overset{\text{geometric}}{=} \beta_{\max}^2 \cdot \left( \left( \frac{\beta_{\min}}{\beta_{\max}} \right)^{2s} - \left( \frac{\beta_{\min}}{\beta_{\max}} \right)^{2t} \right), \quad \bar{\beta}^2 := \beta_{\max}^2 - \beta_{\min}^2, \quad \gamma_t := \frac{\kappa_{t|0}}{\bar{\beta}^2}.$$

$$(68)$$

With them, the conditional base distribution when $f := 0$ can be represented compactly by

$$p^{\text{base}}(X_t | X_0) = \mathcal{N}(X_t; X_0, \kappa_{t|0} I) \tag{69a}$$

$$p^{\text{base}}(X_t | X_0, X_1) = \mathcal{N}(X_t; (1 - \gamma_t) X_0 + \gamma_t X_1, \bar{\beta}^2 \gamma_t (1 - \gamma_t) I) \tag{69b}$$

- The *constant noise schedule* simply sets

$$\sigma_t \overset{\text{constant}}{:=} \sigma. \tag{70}$$

When $f := 0$, the base SDE is effectively a standard Brownian motion whose conditional distributions obey

$$p^{\text{base}}(X_t | X_0) = \mathcal{N}(X_t; X_0, \sigma^2 t I) \tag{71a}$$

$$p^{\text{base}}(X_t | X_0, X_1) = \mathcal{N}(X_t; (1 - t) X_0 + t X_1, \sigma^2 t (1 - t) I) \tag{71b}$$

**Replay buffers $\mathcal{B}_{\text{adj}}$ and $\mathcal{B}_{\text{crt}}$**      Similar to many previous diffusion samplers (Havens et al., 2025; Akhound-Sadegh et al., 2024; Chen et al., 2025), we employ replay buffers $\mathcal{B}$ in computation of both adjoint (14) and corrector (15) matching objectives. Specifically, we rebase the expectation over model samples $p^{u^{(k)}}$ onto a replay buffer $\mathcal{B}$, which stores the most latest $|\mathcal{B}|$ samples. We update the buffer with $N$ new samples every $L$ gradient steps. Note that the use of replay buffers effectively render ASBS a hybrid method between on-policy and off-policy.

**Parametrization of $u_\theta$ and $h_\phi$**     For each energy function, we parametrize the drift $u_\theta(t, x)$ and the corrector $h_\phi(x)$ with two neural networks, $v_\theta(t, x)$ and $v_\phi(t, x)$, of the same architecture.

Specifically, we parametrize the drift as $u_\theta(t, x) := \sigma_t v_\theta(t, x)$, which effectively eliminates the noise schedule "$\sigma_t$" in matching target (see (14)), making it time-invariant for each sampled trajectory. The only exception is the conformer generation task, where we keep the original parametrization $u_\theta(t, x) := v_\theta(t, x)$, which empirically yields better results. On the other hand, since $h_\phi(x)$ is independent of time, we simply set a fixed time input $t = 1$, *i.e.,* $h_\phi(x) := v_\phi(1, x)$.

The specific parametrization $v(t, x)$ employed for each task are detailed below.

- *MW-5*: We consider $v(t, x)$ a standard fully-connected network with 4 layers with 64 hidden features of the following form:

$$\mathtt{output} = \mathtt{layer\_n} \circ \cdots \circ \mathtt{layer\_1} \circ (\mathtt{x\_embed}(x) + \mathtt{t\_embed}(t))$$

- *DW-4, LJ-13, LJ-55*: We consider $v(t, x)$ a Equivariant Graph Neural Network (EGNN; Satorras et al., 2021) with 5 layers and 128 hidden features. The architecture of EGNN is aligned with prior methods (Akhound-Sadegh et al., 2024; Havens et al., 2025).
- *Alanine dipeptide*: We use the same architecture as in MW-5, except with 8 layers with 256 hidden features.
- *Conformer generation*: We consider $v(t, x)$ a similar EGNN used in Adjoint Sampling (Havens et al., 2025), except with 20 layers. Ablation study on the same EGNN architecture can be found in Appendix D.4.

**Clipping $\alpha_{\max}$**     We clip the energy gradient to prevent its maximum norm from exceeding $\alpha_{\max}$.

**Time scaling $\lambda_t$**     Following standard practices for AM objective, we employ a time scaling $\lambda_t$ to improve numerical stability. Note that this does not affect the minimizer of the AM objective. We set $\lambda_t := \frac{1}{\sigma_t^2}$ for all tasks.

**Translation invariance**     For DW-4, LJ-13, LJ-55, and conformer generation tasks, we follow prior methods (Akhound-Sadegh et al., 2024; Havens et al., 2025) by restricting the state space to a zero center-of-mass (ZCOM) subspace and thereby enforcing translation invariance.

For a $n$-particle $k$-dimensional system, *i.e.,* $x = [x_1; \cdots ; x_n]$ where $x_i \in \mathbb{R}^k$, the ZCOM subspace is defined as $\mathcal{X}^{\mathrm{ZCOM}} = \{x \in \mathbb{R}^{nk} : \sum_{i=1}^{n} x_i = 0\}$. Practically, this is achieved by projecting the initial sample $X_0 \sim \mu$, the SDE's noise $\mathrm{d}W_t$, and the energy gradient $\nabla E(\cdot)$ onto $\mathcal{X}^{\mathrm{ZCOM}}$. Note that the output of EGNN is by construction ZCOM.

Formally, the adaption is equivalent to augmenting the SDE with a projection matrix $A \in \mathbb{R}^{nk \times nk}$:

$$\mathrm{d}X_t = \sigma_t A u_t(X_t)\mathrm{d}t + \sigma_t A\mathrm{d}W_t, \quad X_0 = AY_0, \quad Y_0 \sim \mu, \quad A = \left(I_n - \frac{1}{n}\mathbf{1}_n\mathbf{1}_n^\top\right) \otimes I_k, \quad (72)$$

where $\otimes$ is the Kronecker product, $I_n \in \mathbb{R}^{n \times n}$ is an identity matrix, and $\mathbf{1}_n \in \mathbb{R}^n$ is a vector of ones.

**Initialization and alternate procedure**     As ASBS is an instantiation of the IPF algorithm (see Theorem 3.2), it must adhere to the IPF initialization protocol to ensure theoretical convergence to the global solution. Specifically, the IPF initialization can be implemented in two ways

- Initialize with $h_\phi^{(0)} := 0$ and run AM, CM, ... until convergence. We adopt this setup for all tasks.
- Initialize with $u_\theta^{(0)} := 0$ and run CM, AM, ... until convergence. Since $p^{u^{(0)}} = p^{\mathrm{base}}$ in this setup, the optimal corrector at the first CM stage is known analytically:

$$
\begin{aligned}
h^{(1)}(x) &\overset{(15)}{=} \int p_{0|1}^{\mathrm{base}}(y|x)\nabla_x \log p_{1|0}^{\mathrm{base}}(x|y)\mathrm{d}y \\
&= \int \frac{p_{0|1}^{\mathrm{base}}(y|x)}{p_{1|0}^{\mathrm{base}}(x|y)}\nabla_x p_{1|0}^{\mathrm{base}}(x|y)\mathrm{d}y \\
&= \frac{1}{p_1^{\mathrm{base}}(x)}\nabla_x \int p_0^{\mathrm{base}}(y)p_{1|0}^{\mathrm{base}}(x|y)\mathrm{d}y \\
&= \nabla \log p_1^{\mathrm{base}}(x) \qquad\qquad (73)
\end{aligned}
$$

Table 5: Hyperparameters of ASBS for the each task.

| | Synthetic energy functions | | | | Alanine dipeptide | Conformer generation |
|---|---|---|---|---|---|---|
| | MW-5 | DW-4 | LJ-13 | LJ-55 | | |
| $\mu$ | $\mathcal{N}(0,1)$ | $\mu_{\text{harmonic}}$ in (19) with $\alpha=2,2,1$ | | | $\mathcal{N}(0,0.25)$ | $\mu_{\text{harmonic}}$ |
| $\beta_{\min}$ | — | 0.001 | 0.001 | 0.001 | 0.001 | 0.001 |
| $\beta_{\max}$ | — | 1 | 1 | 2 | 0.5 | 1 |
| $\sigma$ | 0.2 | — | — | — | — | — |
| $K$ | 5 | 20 | 15 | 15 | 15 | 3 |
| $M_{\text{adj}}$ | 100 | 200 | 300 | 300 | 4000 | 2500 |
| $M_{\text{crt}}$ | 20 | 20 | 20 | 20 | 2000 | 2000 |
| $N$ | 1000 | 1000 | 1000 | 1000 | 1000 | 128 |
| $L$ | 200 | 100 | 100 | 100 | 100 | 100 |
| $|\mathcal{B}|$ | $10^4$ | $10^4$ | $10^4$ | $10^4$ | $10^4$ | $6.4 \times 10^4$ |
| $\alpha_{\max}$ | — | 100 | 100 | 100 | 100 | 150 |
| $\lambda_t$ | $\frac{1}{\sigma_t^2}$ | $\frac{1}{\sigma_t^2}$ | $\frac{1}{\sigma_t^2}$ | $\frac{1}{\sigma_t^2}$ | $\frac{1}{\sigma_t^2}$ | $\frac{1}{\sigma_t^2}$ |

In practice, we find that the two setups yield similar performance.

**RDKit warm-start** This warm-starts the drift $u_\theta$ using RDKit samples. The procedure is inspired by the fact that (Shi et al., 2023; Liu et al., 2023):

$$
\begin{aligned}
u_t^\star &= \sigma_t \nabla \log \varphi_t \\
&= \arg\min_{u_t} \mathbb{E}_{p_{t,1}^\star} \left[ \| u_t(X_t) - \sigma_t \nabla_{x_t} \log p^{\text{base}}(X_1|X_t) \|^2 \right] \\
&= \arg\min_{u_t} \mathbb{E}_{(X_0,X_1)\sim p_{0,1}^\star, X_t \sim p^{\text{base}}(\cdot|X_0,X_1)} \left[ \| u_t(X_t) - \sigma_t \nabla_{x_t} \log p^{\text{base}}(X_1|X_t) \|^2 \right]. \quad (74)
\end{aligned}
$$

where the last equality is due to

$$
\begin{aligned}
p_{0,t,1}^\star(x,y,z) &\overset{(41)}{=} p_{t,1|0}^{\text{base}}(y,z|x)\hat{\varphi}_0(x)\varphi_1(z) \\
&= p_{t|0,1}^{\text{base}}(y|x,z)p_{1|t}^{\text{base}}(z|y)\hat{\varphi}_0(x)\varphi_1(z) \qquad \text{by Markov property} \\
&\overset{(43)}{=} p_{t|0,1}^{\text{base}}(y|x,z)p_{0,1}^\star(x,z). \quad (75)
\end{aligned}
$$

Equation (74) can be understood as an analogy of (45) for another SB potential $\varphi_t$. In practice, given RDKit samples $X_1 \sim q^{\text{RDKit}}$, we warm-start ASBS by minimizing w.r.t. the following objective:

$$
\begin{aligned}
\mathcal{L}_{\text{warmup}}(\theta) &= \mathbb{E}_{t\sim\mathcal{U}[0,1], X_0\sim\mu, X_1\sim q^{\text{RDKit}}, X_t\sim p^{\text{base}}(\cdot|X_0,X_1)} \left[ \tilde{\lambda}_t \| u_t(X_t) - \sigma_t \nabla_{x_t} \log p^{\text{base}}(X_1|X_t) \|^2 \right] \\
&\overset{(69a)}{=} \mathbb{E}_{t\sim\mathcal{U}[0,1], X_0\sim\mu, X_1\sim q^{\text{RDKit}}, X_t\sim p^{\text{base}}(\cdot|X_0,X_1)} \left[ \tilde{\lambda}_t \| u_t(X_t) - \frac{\sigma_t}{\kappa_{1|t}}(X_1 - X_t) \|^2 \right], \quad (76)
\end{aligned}
$$

where $\kappa_{1|t}$ is defined in (68) for the geometric noise schedule. We set the time scaling $\tilde{\lambda}_t := \sqrt{\frac{\sigma_t}{\kappa_{1|t}}}$.

Note that, unlike AS, the minimizer of (76) does not equal $u^\star$, since $(X_0, X_1) \sim \mu \otimes q^{\text{RDKit}} \neq p_{0,1}^\star$ are sampled independently.

# D Experiment Details

## D.1 Synthetic Energy Functions

### D.1.1 Energy functions

In this section, we provide the exact setup for our synthetic energy experiments in Table 2. We consider four synthetic energy functions that have been widely used in recent literature to benchmark sampling and generative algorithms: MW-5, DW-4, LJ-13, and LJ-55.

**MW-5** The MW-5 (Many-Well in 5D) energy is a 5-particle 1D system adopted from Chen et al. (2025), where $x = [x_1; \cdots ; x_5] \in \mathbb{R}^5$ with $x_i \in \mathbb{R}$, . The energy function is defined as follows:

$$E(x) = \sum_{i=1}^{5} (x_i^2 - \delta)^2 \tag{77}$$

where we set $\delta = 4$. This creates distinct modes centered at combinations of $\pm\sqrt{\delta}$ in each of the $d$ dimensions.

**DW-4** The DW-4 (Double-Well for 4 particles in 2D) energy is a physically motivated pairwise potential originally proposed in Köhler et al. (2020) and subsequently used in Akhound-Sadegh et al. (2024); Havens et al. (2025). It defines a system of four particles, each living in $\mathbb{R}^2$, leading to an 8D state vector $x = [x_1; x_2; x_3; x_4] \in \mathbb{R}^8$ with $x_i \in \mathbb{R}^2$. The energy function reads

$$E(x) = \exp\left[\frac{1}{2\tau} \sum_{i<j} \left(a(d_{ij} - d_0) + b(d_{ij} - d_0)^2 + c(d_{ij} - d_0)^4\right)\right], \tag{78}$$

where $d_{ij} = \|x_i - x_j\|_2$ is the Euclidean distance between particles $i$ and $j$. We follow the standard configuration with $a = 0$, $b = -4$, $c = 0.9$, $d_0 = 1$, and temperature $\tau = 1$.

**LJ-13 and LJ-55** The Lennard-Jones (LJ) potentials are classical intermolecular potentials commonly used in physics to model atomic interactions. These are defined for a system of $n$ particles in 3D space, with $x = [x_1; \ldots; x_n] \in \mathbb{R}^{3n}$ and $x_i \in \mathbb{R}^3$. The index following "LJ-" indicates the number of particles (e.g., 13 or 55). The unnormalized energy function takes the form:

$$E(x) = \frac{\epsilon}{2\tau} \sum_{i<j} \left[\left(\frac{r_m}{d_{ij}}\right)^6 - \left(\frac{r_m}{d_{ij}}\right)^{12}\right] + \frac{c}{2} \sum_{i} \|x_i - C(x)\|^2, \tag{79}$$

where $d_{ij} = \|x_i - x_j\|_2$ is the pairwise distance and $C(x)$ denotes the center of mass of the particles. We use the parameter values $r_m = 1$, $\epsilon = 1$, $c = 0.5$, and $\tau = 1$, following prior work. The LJ-13 and LJ-55 systems correspond to 39D and 165D, respectively.

### D.1.2 Baselines

Here, we outline the procedure used to obtain the values reported in Table 2 for the baseline methods.

For PIS (Zhang and Chen, 2022), DDS (Vargas et al., 2023), and LV-PIS (Richter and Berner, 2024), iDEM (Akhound-Sadegh et al., 2024), and AS (Havens et al., 2025), we reuse the values reported in AS (Havens et al., 2025, Table 1) for DW-4, LJ-13, and LJ-55 energy functions. As for MW-5, which is not included in AS, we run iDEM using their official implementation and the rest of baseline methods using our own implementation in PyTorch (Paszke et al., 2019). We were unable to obtain reportable results for LV-PIS and iDEM on this energy function.

For PDDS (Phillips et al., 2024) and SCLD (Chen et al., 2025), we run their official implementations in JAX (Bradbury et al., 2018) using the default hyperparameter settings specified for the Log-Gaussian Cox Process experiment in their respective papers. To enhance stability and convergence on synthetic energy functions, we tune the gradient clipping parameters. For PDDS, we apply clipping to the gradient of the energy function. For SCLD, we clip both the energy gradient and the Langevin norm. In both cases, the clipping magnitude is selected from the set $\{1, 10, 100, 1000\}$ based on the best validation performance. Training is performed for 100,000 iterations across all runs. For SCLD, we use subtrajectory splitting with the default value of 4, so that it does not degenerate to CMCD (Vargas et al., 2024). In practice, we find that using subtrajectories yields better results.

### D.1.3 Evaluation Metrics

In this subsection, we outline the evaluation criteria used to quantitatively assess the quality of samples generated from synthetic energy functions. We employ three primary metrics: Sinkhorn distance, geometric $\mathcal{W}_2$, and energy $\mathcal{W}_2$, each designed to capture different aspects of distributional similarity between generated and ground truth samples.

**Sinkhorn distance** To evaluate the similarity between the empirical distributions of generated and reference samples, we compute the Sinkhorn distance using the entropy-regularized optimal transport

formulation (Peyré and Cuturi, 2019), following the implementation of Blessing et al. (2024) and Chen et al. (2025). The Sinkhorn regularization coefficient is set to $10^{-3}$ throughout. We use 2,000 samples from both the generated and ground truth distributions to compute the metric.

**Geometric $\mathcal{W}_2$** For DW and LJ tasks, the potential energy functions—and consequently, the sample distributions—exhibit invariance to both particle permutations and rigid transformations such as rotations and reflections. To appropriately account for these symmetries, we employ the geometric $\mathcal{W}_2$ distance as defined by Akhound-Sadegh et al. (2024) and Havens et al. (2025). Formally, the 2-Wasserstein distance is computed as:

$$\mathcal{W}_2^2(\hat{\nu}, \nu) = \inf_{\pi \in \Pi(\hat{\nu}, \nu)} \int D(x, y)^2 \, \pi(x, y) \, dxdy, \tag{80}$$

where $\Pi(\hat{\nu}, \nu)$ denotes the set of joint couplings with prescribed marginals $\hat{\nu}$ (generated) and $\nu$ (ground truth), and $D(x, y)$ is a symmetry-aware distance between samples defined as:

$$D(x, y) = \min_{R \in O(s), \, P \in S(n)} \|x - (R \otimes P)y\|_2. \tag{81}$$

Here, $O(s)$ denotes the group of orthogonal transformations in $s$ spatial dimensions (rotations and reflections), and $S(n)$ represents the symmetric group over $n$ particles. As exact minimization over these symmetry groups is computationally infeasible, we adopt the approximation scheme of Köhler et al. (2020). We use 2000 samples from each generated and ground truth distribution to compute the metric.

**Energy $\mathcal{W}_2$** To evaluate fidelity with respect to the target energy landscape, we also compute the 2-Wasserstein distance between the energy values of generated samples and those of ground truth samples. For each target distribution, we generate 2,000 samples from both the model and the reference, and compare their respective energy histograms. This scalar-based Wasserstein metric serves as a proxy for how well the generative model captures the energy histogram of the target distribution.

## D.2 Alanine dipeptide

**Benchmark description** We adopt the experiment setup primarily from (Midgley et al., 2023). Given a configuration of alanine dipeptide, which consists of 22 particles in 3D, *i.e.,* $x = [x_1; \cdots; x_{22}] \in \mathbb{R}^{66}$ where $x_i \in \mathbb{R}^3$, we apply the same coordinate transform $\mathcal{T}$ proposed by Midgley et al. (2023). This coordinate transform maps the Cartesian coordinates to internal coordinates, $\mathcal{T}(x) =: z \in \mathbb{R}^{60}$, which include bond lengths, bond angles, and dihedral angles (Stimper et al., 2022). This process effectively removes six degrees of freedom—three for translation and three for rotation—thereby enforcing structural invariance. Non-angular coordinates are further normalized using samples with minimal energies. We refer readers to (Midgley et al., 2023, Appendix F.1) for further details. Note that the internal coordinate transformation is bijective. Hence, we can compute the energy via

$$E(x) = E(\mathcal{T}^{-1}(z)) \tag{82}$$

**Evaluation and baselines** For each sample $x = \mathcal{T}^{-1}(z) \in \mathbb{R}^{66}$, we extract five torsion angles, including the backbone angles $\phi$, $\psi$ and methyl rotation angles $\gamma_1, \gamma_2, \gamma_3$. We report two divergence metrics with respect to the ground-truth distribution, which contains $10^7$ samples simulated by Molecular Dynamics. We implement the baseline methods, including PIS (Zhang and Chen, 2022), DDS (Vargas et al., 2023), AS (Havens et al., 2025), using PyTorch (Paszke et al., 2019).

For the KL divergences, we adopt setup from (Wu et al., 2020) and compute the divergence of the ground-truth marginal to model marginal for each torsion angle:

$$D_{\mathrm{KL}}(p^\star(\cdot)||p^{u_\theta}(\cdot)) \approx \sum P^\star(\cdot) \log \frac{P^\star(\cdot) + \epsilon}{P^{u_\theta}(\cdot) + \epsilon}, \qquad \epsilon = 10^{-5}, \tag{83}$$

where $P^\star$ and $P^{u_\theta}$ are histograms of $10^7$ samples, discretized between $[-\pi, \pi]$ with 200 intervals.

For the Wasserstein-2 distance, we use the Geometric $\mathcal{W}_2$ in (80), where each sample is now in 2D, $x = [\phi, \psi] \in \mathbb{R}^2$. Due to the high computational cost, we compute the value using a subset of $10^4$ samples from the test set ground-truth samples, which is fixed for all methods.

Finally, both Ramachandran plots in Figure 5 are generated using $10^7$ samples.

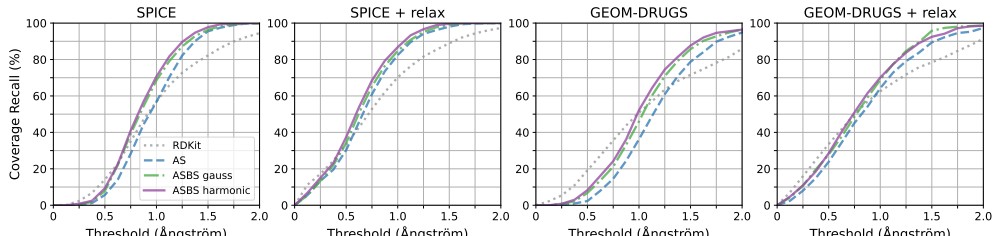

Figure 8: Ablation study on full recall coverage curves (without RDKit warm-start) using the same EGNN architecture as in AS (Havens et al., 2025). Note that Table 6 reports the values at the thresholds **1.0Å** and **1.25Å**.

### D.3 Amortized conformer generation

In this subsection, we provide some context for the experimental results found in Table 4 regarding the generation of conformers.

**Benchmark description** Conformers are atomic representations of molecules in cartesian space with their constituent atoms arranged into local minima on the potential energy surface. Molecules are defined to be a graph of atoms (nodes) connected by bonds (edges); conformers are geometric realizations of that molecule. Torsion angles, or rotatable bonds, are particularly important degrees of freedom for defining conformations since bond lengths and bond angles are typically much more stable due to a high sensitivity to perturbations. It is common to consider bond lengths and bond angles fixed, while the torsional degrees of freedom define the conformer.

The task in this benchmark is to take a representation of the molecular graph, usually a SMILES string (Weininger, 1988), and comprehensively sample the conformational configuration space. In flexible molecules, there can be a large number of conformers with many separated modes in a $3n - 6$ dimensional space. (Where $n$ represents the number of atoms and 6 comes from the irrelevance of rotation and translation of the conformer.) We quantify the notion of comprehensively sampling the space by comparing generated structures to a set of conformers sampled using expensive, standard search techniques (Pracht et al., 2024) that were further relaxed using extremely precise density function theory-based, quantum chemistry methods (Neese, 2012; Levine et al., 2025). A detailed description of this benchmark can be found in its source (Havens et al., 2025, Appendix F.).

**Evaluation and baselines** The method of comparison between proposed structure and reference conformer is to use RDKit's (Landrum, 2006) implementation of *Root Mean Squared Displacement* (RMSD), a measure of distance between atomic structures that is invariant to translation and rotation. We set a threshold RMSD for two structures to match and computed the Recall Coverage and Recall Average Minimum RMSD (AMR). The experiment was performed with both generated structures and with generated structures after a so-called relaxation, i.e. geometry optimization of energy, using eSEN (Fu et al., 2025). The equations for computing these metrics are:

$$\text{COV-R}(\delta) := \frac{1}{L} \left| \{ l \in \{1, \dots, L\} : \exists k \in \{1, \dots, K\}, \quad \text{RMSD}(C_k, C_l^*) < \delta \} \right| \quad (84)$$

$$\text{AMR-R} := \frac{1}{L} \sum_{l \in \{1, \dots, L\}} \min_{k \in \{1, \dots, K\}} \text{RMSD}(C_k, C_l^*) \quad (85)$$

where $\delta = 0.75$ Å is the coverage threshold, $L = \max(L', 128)$, where $L'$ is the number of reference conformers, $K = 2L$, and let $\{C_l^*\}_{l \in [1,L]}$ and $\{C_k\}_{k \in [1,K]}$ be the sets of ground truth and generated conformers respectively. We capped the reference conformers per molecule at 512 in COV-R.

The values for the baselines are adopted from AS (Havens et al., 2025).

### D.4 Additional Experiments and Discussions

**Ablation study between AS and ASBS using the same EGNN** For the amortized conformer generation task in Table 4, we use an EGNN architecture with 20 layers, whereas AS employs the same architecture with 12 layers. In Table 6, we report the results of ASBS using the same 12-layer

Table 6: Ablation study on amortized conformer generation using the same EGNN architecture as in AS (Havens et al., 2025). We report the recall at the thresholds **1.0Å** and **1.25Å**, where the latter was reported in AS.

| | without relaxation | | | | with relaxation | | | |
| --- | --- | --- | --- | --- | --- | --- | --- | --- |
| | SPICE | | GEOM-DRUGS | | SPICE | | GEOM-DRUGS | |
| Method | Coverage ↑ | AMR ↓ | Coverage ↑ | AMR ↓ | Coverage ↑ | AMR ↓ | Coverage ↑ | AMR ↓ |
| **Threshold 1.0Å** | | | | | | | | |
| RDKit ETKDG (Riniker and Landrum, 2015) | $56.94_{\pm35.82}$ | $1.04_{\pm0.52}$ | $50.81_{\pm34.69}$ | $1.15_{\pm0.61}$ | $70.21_{\pm31.70}$ | $0.79_{\pm0.44}$ | $62.55_{\pm31.67}$ | $0.93_{\pm0.53}$ |
| AS (Havens et al., 2025) | $56.75_{\pm38.15}$ | $0.96_{\pm0.26}$ | $36.23_{\pm33.42}$ | $1.20_{\pm0.43}$ | $82.41_{\pm25.85}$ | $0.68_{\pm0.28}$ | $64.26_{\pm34.57}$ | $0.89_{\pm0.45}$ |
| ASBS w/ Gaussian prior (**Ours**) | $68.61_{\pm33.48}$ | $0.88_{\pm0.25}$ | $46.03_{\pm35.99}$ | $1.08_{\pm0.36}$ | $84.77_{\pm22.65}$ | $0.64_{\pm0.25}$ | $68.83_{\pm31.53}$ | $0.80_{\pm0.37}$ |
| ASBS w/ Harmonic prior (**Ours**) | $70.70_{\pm33.21}$ | $0.86_{\pm0.24}$ | $52.19_{\pm35.93}$ | $1.05_{\pm0.41}$ | $86.79_{\pm22.86}$ | $0.61_{\pm0.24}$ | $70.08_{\pm31.60}$ | $0.80_{\pm0.37}$ |
| AS +RDKit warmup (Havens et al., 2025) | $72.21_{\pm30.22}$ | $0.84_{\pm0.24}$ | $52.19_{\pm35.20}$ | $1.02_{\pm0.34}$ | $87.84_{\pm19.20}$ | $0.60_{\pm0.23}$ | $73.88_{\pm28.63}$ | $0.76_{\pm0.34}$ |
| ASBS +RDKit warmup (**Ours**) | $74.29_{\pm31.25}$ | $0.82_{\pm0.24}$ | $55.88_{\pm36.51}$ | $0.98_{\pm0.34}$ | $87.25_{\pm20.77}$ | $0.60_{\pm0.24}$ | $74.11_{\pm30.16}$ | $0.75_{\pm0.34}$ |
| **Threshold 1.25Å** | | | | | | | | |
| RDKit ETKDG (Riniker and Landrum, 2015) | $72.74_{\pm33.18}$ | $1.04_{\pm0.52}$ | $63.51_{\pm34.74}$ | $1.15_{\pm0.61}$ | $81.61_{\pm27.58}$ | $0.79_{\pm0.44}$ | $71.72_{\pm29.73}$ | $0.93_{\pm0.53}$ |
| AS (Havens et al., 2025) | $82.22_{\pm25.72}$ | $0.96_{\pm0.26}$ | $60.93_{\pm35.15}$ | $1.20_{\pm0.43}$ | $94.10_{\pm15.67}$ | $0.68_{\pm0.28}$ | $79.08_{\pm29.44}$ | $0.89_{\pm0.45}$ |
| ASBS w/ Gaussian prior (**Ours**) | $87.20_{\pm21.88}$ | $0.88_{\pm0.25}$ | $70.86_{\pm31.98}$ | $1.08_{\pm0.36}$ | $95.19_{\pm10.29}$ | $0.64_{\pm0.25}$ | $84.66_{\pm25.03}$ | $0.80_{\pm0.37}$ |
| ASBS w/ Harmonic prior (**Ours**) | $89.66_{\pm19.42}$ | $0.86_{\pm0.24}$ | $74.50_{\pm32.32}$ | $1.05_{\pm0.41}$ | $96.64_{\pm10.15}$ | $0.61_{\pm0.24}$ | $83.76_{\pm24.77}$ | $0.80_{\pm0.37}$ |
| AS +RDKit warmup (Havens et al., 2025) | $89.42_{\pm17.48}$ | $0.84_{\pm0.24}$ | $72.98_{\pm30.82}$ | $1.02_{\pm0.34}$ | $96.65_{\pm7.51}$ | $0.60_{\pm0.23}$ | $87.01_{\pm22.79}$ | $0.76_{\pm0.34}$ |
| ASBS +RDKit warmup (**Ours**) | $90.85_{\pm17.74}$ | $0.82_{\pm0.24}$ | $77.86_{\pm30.37}$ | $0.98_{\pm0.34}$ | $97.28_{\pm6.55}$ | $0.60_{\pm0.24}$ | $87.81_{\pm22.75}$ | $0.75_{\pm0.34}$ |

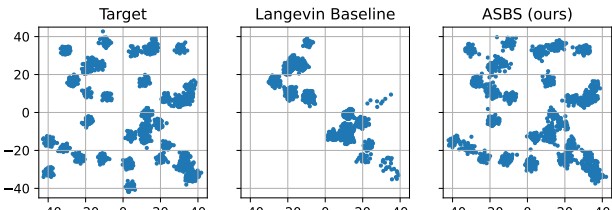

Figure 9: Compared to vanilla Langevin baseline, our ASBS—instantiated with a standard uni-variance Gaussian—is able to identify almost all modes without any prior knowledge of where the target modes were located.

EGNN as AS. Notably, our ASBS consistently outperforms AS on all metrics across all setups, except the coverage for GEOM-DRUGS with relaxation and RDKit warm-start, where ASBS falls slightly behind AS by only 1.0%. Finally, Figure 8 reports the full recall coverage curves that reproduce Table 4.

**Ability of ASBS in finding modes**     We conduct additional experiments on the 40-mode GMM in 2D. Specifically, we instantiate ASBS with a uni-variance Gaussian source distribution centered at zero, effectively assuming no prior knowledge of the target modes, as the initial distribution does not coincide with any target modes. We also run a vanilla Langevin baseline for 1 million steps, starting from the same source distribution.

Figure 9 represents the quantitative results. Notably, ASBS is able to identify almost all modes. In contrast, the vanilla Langevin baseline appears to suffer from a slow mixing rate, recovering less than half of the total modes even after 1 million steps. We highlight this distinction as an advantage of constructing diffusion samplers from the stochastic control and Schrödinger Bridge frameworks, which allows theoretical convergence to target distribution within a finite horizon. Finally, we believe that with proper tuning of the ASBS noise schedule, its performance can be further enhanced.

**Discussion on important weights**     Finally, we discuss the potential integration of ASBS with importance weights, emphasizing that our theoretical and algorithmic frameworks do not preclude the use of importance weights to further enhance performance or robustness.

Formally, the importance weights over model path $X \sim p^u$ admit the following representation:

$$w(X) := \frac{\mathrm{d}p^\star(X)}{\mathrm{d}p^u(X)} = \exp\left( \int_0^1 -\tfrac{1}{2}\|u_t(X_t)\|^2 \mathrm{d}t - \int_0^1 u_t(X_t) \cdot \mathrm{d}W_t - \log\frac{\hat{\varphi}_1(X_1)}{\nu(X_1)} + \log\frac{\hat{\varphi}_0(X_0)}{\mu(X_0)} \right),$$
(86)

which can be obtained from (59) by setting $\bar{h} := \hat{\varphi}_1$ so that $q^{\bar{h}} = p^\star$ is the optimal distribution of SB.

Note that when the source distribution degenerates to the Dirac delta $\mu(X_0) = \delta_0(X_0)$, the last term $\log \frac{\hat{\varphi}_0(X_0)}{\mu(X_0)}$ becomes a constant and—as discussed in Section 3.2—$\hat{\varphi}_1 = p_1^{\text{base}}$, thereby recovering the weights used in prior SOC-based methods (Zhang and Chen, 2022; Havens et al., 2025).

Equation (86) is also a more concise representation than the one derived in (Richter and Berner, 2024), by recognizing the following relation through the application of Ito Lemma (46) to $\log \hat{\varphi}_t(X_t)$:

$$\frac{\log \hat{\varphi}_1(X_1)}{\log \hat{\varphi}_0(X_0)} = \int_0^1 \left[ \tfrac{1}{2} \| v_t(X_t) \|^2 + (u_t \cdot v_t)(X_t) + \nabla \cdot (\sigma_t v_t(X_t) - f_t(X_t)) \right] \mathrm{d}t + \int_0^1 v_t(X_t) \cdot \mathrm{d}W_t, \tag{87}$$

where we shorthand $v_t(x) := \sigma_t \nabla \log \hat{\varphi}_t(x)$.

Estimating the weight in (86) requires knowing the ratios $\frac{\hat{\varphi}_1(x)}{\nu(x)}$ and $\frac{\hat{\varphi}_0(x)}{\mu(x)}$, which are not immediately available with the current parametrization, $u_\theta(t, x) \approx \sigma_t \nabla \log \varphi_t(x)$ and $h_\phi(x) \approx \nabla \log \hat{\varphi}_1(x)$. One accommodation is to reparametrize the functions with potential network $v(t, x) : [0, 1] \times \mathcal{X} \to \mathbb{R}$,

$$u_\theta(t, x) := \sigma_t \nabla v_\theta(t, x), \qquad h_\phi(x) := \nabla v_\phi(1, x) \tag{88}$$

and then regress their gradients onto the adjoint and corrector targets. With that, the logarithmic ratios can be easily estimated:

$$\log \frac{\hat{\varphi}_1(x)}{\nu(x)} = v_\phi(1, x) + E(x), \qquad \log \frac{\hat{\varphi}_0(x)}{\mu(x)} \overset{(42)}{=} - \log \varphi_0(x) = v_\theta(0, x). \tag{89}$$

A more detailed investigation of this importance sampling scheme is left for future work.

