# OpenReview forum: "Adjoint Schrödinger Bridge Sampler"
_NeurIPS.cc/2025/Conference — NeurIPS 2025 oral_

### Official Review · Reviewer_NUeM · 2025-06-28

**Clarity:** 4
**Significance:** 3
**Originality:** 3
**Rating:** 6
**Confidence:** 3

**Summary:**

This paper considers how to modify a reference SDE in order to sample from an unnormalized density. Like prior literature, it proposes to correct the SDE with an additive drift that should be learnt (Eq 2). Overall, there are two ways of learning the drift in the literature:

*(i) This drift is learnt by minimizing a trajectory-level cost (Eq 4) and a general terminal cost*

Computationally, this problem can be efficiently solved using so-called adjoint sampling (Eq 14).

Theoretically, the resulting SDE does *not* sample the target density unless some additional, restrictive assumptions are made (memoryless schedule and Dirac initialization).

*(ii) The drift is learnt by minimizing a trajectory-level cost (Eq 3) under a terminal condition*

Computationally, this classical problem called Schrodinger Bridge can be solved using an Iterative Proportional Fitting algorithm.

Theoretically, the resulting SDE *does* sample the target density.

The authors identify a special terminal cost (Eq 9) so that (i) and (ii) are theoretically equivalent, and in particular the resulting SDE samples from the target density. Because the special terminal cost depends on an unknown quantity, the authors propose to learn it on-the-go (Eq 15). This way, the authors aim to use the computational efficiency of (i) to solve the problem in (ii).

**Questions:**

Q1. The authors propose a nested optimization problem, where the outer loop updates follow Eq 14 and the inner loop updates follow Eq 15. How brittle is the interplay between these two updates? For example, in the inner loop which approximates Eq 15, how many iterations of the optimizer are required to get stable results?

Q2. Could the authors illustrate their algorithm's ability to find the modes, by illustrating it on the synthetic 40-mode Gaussian mixture in 2D from Figure 2 of [1]? Please use an initial distribution that has the same density as one of the modes of the mixture (so as to not assume that the user knows where the other modes are).

The resulting plot could be reported in the Appendix and could compare only ASBS and a vanilla Langevin to get a visual understanding of how the authors' algorithm deals with the "search problem" (finding the target modes) as compared to a basic local exploration algorithm (Langevin).

Here is an additional discussion of why this plot would be useful. While the results on molecular tasks are impressive, they do not provide a visual understanding of which modes are found by the algorithm. For example, is it those modes that are closest to the initial distribution that are found by ASBS? Similarly, Figure 2, while illustrative, is an example where the target distribution is symmetric and the initial distribution is initialized in the barycenter of the target modes, so even a Langevin process would be an efficient sampler.

**Ethical Concerns:**

["NO or VERY MINOR ethics concerns only"]

**Final Justification:**

I maintain my recommendation for a strong acceptance.

The theory is well-presented and impactful. The authors satisfactorily conducted experiments that answered the questions I had in my review, on the stability of their optimization scheme and on the mode-finding ability of their algorithm.

I have also read the comments by the other reviewers which conclude overall that the theoretical contributions are worthwhile for the community, while the experiments are satisfactory but could be improved, which the authors did in their rebuttal.

**Limitations:**

Yes.

**Paper Formatting Concerns:**

None.

**Quality:**

3

**Strengths And Weaknesses:**

## Strengths

The paper is very clearly written and proposes to alternate between adjoint sampling updates (Eq 14) and corrective updates (Eq 15) to learn a corrective drift to an SDE for sampling from an unnormalized density. The alternate optimization scheme (Eqs 14-15) is connected to the classical Iterative Proportional Fitting algorithm which is appreciated. The benchmarks with other diffusion samplers is are satisfactory and the experiments on synthetic and molecular problems are appreciated.

## Weaknesses

One of the main difficulties in sampling from an unnormalized density is finding the modes. It would be nice to visually inspect the algorithm's ability to find the modes as in [1]: please see Q2 for a more detailed request.

It is also unclear how the ASBS algorithm's ability to find modes scales as a function of the dimensionality and of the number of modes.

[1] L. Illing Midgley, V. Stimper, G. N. C. Simm, B. Scho ̈lkopf, and J. Miguel Hernandez-Lobato. Flow annealed importance sampling bootstrap. In Interna- tional Conference on Learning Representations (ICLR), 2023.

---

> ### Author Rebuttal · Authors · 2025-07-31
>
> We thank the reviewer for raising all valuable comments and the comprehensive summary. We are excited that the reviewer identified the novelty of technical contributions and connection to the Iterative Proportional Fitting algorithm, appreciated the well-written presentation, and acknowledged our strong empirical results. We believe ASBS takes a significant step toward scalable diffusion samplers with flexible model design. We try our best to resolve all raised concerns in the responses below.
>
>
> ---
>
> **1. Additional experiments on mode searching**
>
> - We thank the reviewer for raising the comments. Following the reviewer’s suggestion, we conduct additional experiments on the 40-mode GMM in 2D from [1]. Specifically, we instantiate ASBS with a uni-variance Gaussian source distribution centered at zero, effectively assuming that users have no prior knowledge of the target modes, as the initial distribution does not coincide with any target modes. We also run a vanilla Langevin baseline for 1 million steps, starting from the same source distribution.
>
>
> - The table below presents the count of identified modes out of a total of 40. Notably, ASBS is able to identify almost all modes. In contrast, the vanilla Langevin baseline appears to suffer from a slow mixing rate, recovering less than half of the total modes even after 1 million steps. We highlight this distinction as an advantage of constructing diffusion samplers from the stochastic control and Schrödinger Bridge frameworks, which allows theoretical convergence to target distribution within a finite horizon. Finally, we believe that with proper tuning of the ASBS noise schedule, its performance can be further enhanced.
>
>   |  | # of identified modes out of 40 |
>   |---|---|
>   | Vanilla Langevin | 15/40 |
>   | ASBS | 38/40 |
>
>
> - We kindly note that, in accordance with this year’s NeurIPS policy, we are unable to provide any external link to images or updating the manuscript. However, we believe these are valuable discussions and will include the figures in the revision. We thank the reviewer again for bringing up the topic.
>
> ---
>
> **2. Clarification on nested optimization (Eqs 14 & 15)**
>
> - Regarding the sensitivity of the interplay between Adjoint Matching (Eq. 14) and Corrector Matching (Eq. 15), we observe in practice that both optimizations exhibit stable convergence due to their formulation as simple regression (i.e., matching) objectives. While exact convergence to the minima of Eqs. 14 and 15 is required for global convergence (see Thm 3.2), we found that early termination with a fixed number of iterations did not cause training to diverge or become unstable in any of the tasks considered.
>
>
> - In practice, the Corrector Matching (Eq 15) often requires significantly fewer iterations to converge compared to the Adjoint Matching (Eq 14). Specifically, the number of iterations for CM ($M_\text{crt}$) are typically 2 to 10 times fewer than for AM ($M_\text{adj}$). We refer the reviewer to Table 5 in Appendix for the exact values of $M_\text{crt}$ and $M_\text{adj}$ on each task.
>
>
> ---
>
> [1] Flow Annealed Importance Sampling Bootstrap
> [2] Non-Equilibrium Transport Sampler

---

> > ### Comment · Reviewer_NUeM · 2025-08-02
> > **Acknowledgement**
> >
> > Thank you authors: this satisfactorily answers my questions. I suggest these two points be included in the final version of the paper.
> >
> > I am happy to maintain my recommendation for a strong acceptance.

---

### Official Review · Reviewer_jVAy · 2025-07-03

**Clarity:** 4
**Significance:** 3
**Originality:** 4
**Rating:** 5
**Confidence:** 4

**Summary:**

This paper attempts to tackle the problem of sampling from a Boltzmann distribution.  They do so by building on the recent Adjoint Sampling work, noting that the memoryless condition -- an issue first recognized in the Adjoint Matching paper -- may be solved by employing a stochastic optimal control approach to the Schrödinger Bridge problem.  The authors finally demonstrate ASBS's efficacy on a number of sampling tasks.

**Questions:**

1. How does ASBS perform compared to the improved RDKit baseline in the conformer generation task?

2. Can ASBS scale up to any of the even larger scale experimental settings I described in the strengths/weaknesses section?

**Ethical Concerns:**

["NO or VERY MINOR ethics concerns only"]

**Final Justification:**

I believe this paper should be accepted to NeurIPS.  The theory is interesting and experimental results good enough that I believe it merits presentation at the conference.  I recommend a 5.

**Limitations:**

Yes

**Quality:**

4

**Strengths And Weaknesses:**

I found the paper to be quite well written, that the theoretical results -- especially that the memoryless condition can be relaxed via the proposed SOC approach -- were significant, and that the experimental results showed ASBS's efficacy.  As such, I believe the paper merits an acceptance.

I don't have many notes or questions regarding the theoretical section of the paper as I found the theoretical results to be satisfactory.  However, I have a few concerns regarding the paper's experimental validation.  Further, I feel that while ASBS performed well in the presented experimental settings, the experimental settings considered in the paper are not ones which significantly push the boundary on amortized samplers (e.g., ALDP in internal coordinates and the various synthetic energy functions have been solved for some time now).  I already feel the paper merits acceptance, but if the authors were to include further experiments showing that ASBS scales well enough to tackle problems like alanine tripeptide or larger molecules in, e.g., cartesian coordinates (a more challenging task due to how internal coordinates allow one to bypass problems with very sharp energy functions) I would be happy to raise my score to a strong accept.  Note that prior work (albeit it is a Boltzmann generator) in [1] have been able to successfully to these sorts of problems, so it would be interesting to see whether ASBS could scale to this.

As a side note, I believe that the RDKit baseline is not representative of the actual state-of-the-art performance of RDKit based conformer generation, see [2] which achieves superior performance to most deep learning based conformation sampling methods using some different settings of RDKit than the ETKDG baseline used in this paper.

References:

[1] Tan, Charlie B., et al. "Scalable equilibrium sampling with sequential boltzmann generators." ICML (2025).

[2] Zhou, Gengmo, et al. "Do deep learning methods really perform better in molecular conformation generation?." arXiv (2023).

---

> ### Author Rebuttal · Authors · 2025-07-31
>
> We thank the reviewer for raising all valuable comments. We are excited that the reviewer identified the novelty of technical contributions in relaxing the memoryless condition, appreciated the well-written presentation, and acknowledged our strong empirical results. We believe ASBS takes a significant step toward scalable diffusion samplers with flexible model design. We try our best to resolve all raised concerns in the responses below.
>
> ---
>
> **1. Additional experiments on ALDP**
>
> - We first thank the reviewer for raising the comments. The experiments for ALDP (Table 3 & Fig 5) were conducted in internal coordinates mainly to maintain consistency with prior SOC-based diffusion samplers, specifically PIS [4], to which our ASBS sampler also belongs. That being said, our ASBS can indeed be run on the full Cartesian coordinates for ALDP.
>
>
> - In the table below, we report preliminary results for ASBS (using harmonic source) trained from scratch in full Cartesian coordinates, without relying on any explicit target or MCMC samples. Similar to Table 3, we report the KL divergences for the 1D marginal across five torsion angles, $(\phi, \psi, \gamma_1, \gamma_2, \gamma_3)$, as well as the Wasserstein-2 on joint distribution of $(\phi, \psi)$. The latter serves as a quantitative measure of the Ramachandran plot. We kindly note that, in accordance with this year’s NeurIPS policy, we are unable to provide any external link to images or updating the manuscript.
>
>
>   |  | $\phi$ | $\psi$ | $\gamma_1$ | $\gamma_2$ | $\gamma_3$ | $(\phi,\psi)$ |
>   |---|---|---|---|---|---|---|
>   | ASBS (Cartesian) | 0.78 | 0.54 | 0.20 | 0.11 | 0.11 | 1.30 |
>
>
> - Our preliminary results imply that ASBS can generate samples with reasonable structures, with Ramachandran plots that match those of prior SOC-based diffusion samplers in internal coordinates (see Table 3). Notably, ASBS achieves this without using any explicit target samples during training, relying solely on the energy function of ALDP. This contrasts with previous attempts on ALDP in Cartesian coordinates—such as [1,3] mentioned by the reviewer and Reviewer 5V2q—which often require explicit samples from or near the target distribution. We emphasize this distinction as a key advantage of adjoint-based diffusion samplers, which we further detail in the second section.
>
>
> - We would like to highlight that the amortized conformer generation task (Table 4) already includes larger molecules, with 50-65 atoms in the test sets of SPICE and GEOM-DRUG, respectively, whereas the suggested alanine tripeptide consists of 33 atoms. Given ASBS's strong performance in amortized conformer generation and its capability to generate the Boltzmann distribution of ALDP in Cartesian spaces, we believe ASBS has great potential for tackling larger molecules. Scaling ASBS for amortized Boltzmann generators would be a promising direction worthy of its own in-depth investigation.
>
> ---
>
> **2. Discussion on the new RDKit baseline**
>
> - We thank the reviewer for referring us to the new RDKit baseline [2], which post-processes RDKit samples using a K-means clustering algorithm. Following the reviewer’s suggestion, we applied their method to our GEOM-DRUG test set and found that the recall converges at the same 0.75Å threshold, as reported in Table 4, to approximately 54.4%. This performance is comparable to ASBS with a harmonic prior and relaxation post-processing.
>
>
> - We note, however, that [2] appears to generate a much larger number of conformers than the evaluation pipeline used in Table 4. That said, we believe that ASBS could similarly benefit from clustering post-processing by generating a comparable number of conformers. Exploring such post-processing techniques, which have proven effective in prior work, would be an interesting direction for future research. We thank the reviewer for highlighting this opportunity.
>
> ---
>
> [1] Scalable Equilibrium Sampling with Sequential BG
> [2] Do DL Methods Really Perform Better in Molecular Conformation Generation
> [3] Temperature-Annealed BG
> [4] Path Integral Sampler

---

> > ### Comment · Reviewer_jVAy · 2025-08-04
> >
> > I thank the authors for their thorough response to my questions.  I am happy with the provided ALDP results on cartesian coordinates and am appreciative of their efforts on adding the additional RDKit baseline.  I briefly want to highlight that while indeed their SPICE and GEOM-DRUG experiments are impressive, my original intent was to say that I was looking for additional experimental results on larger scale molecules achieved _without_ the use of training data.  Scaling amortized samplers without the use of additional training data is the one of the next frontiers for sampling methods as it is significantly more difficult and unsolved. It was results on this end which might push me towards a higher score.  Still, I maintain that the results in the paper as well as those added during rebuttal are significant enough to merit acceptance and as such I keep my score of 5.

---

> > > ### Author Response · Authors · 2025-08-04
> > >
> > > We thank the reviewer for the reply and are excited that the reviewer appreciated our additional results. We provide additional clarifications below.
> > >
> > > ---
> > >
> > > - We would first like to emphasize that **none of our experiments use training data** (i.e., explicit samples from ground-truth target distribution). This includes the newly conducted ALDP experiments on Cartesian coordinates, where we trained an ASBS sampler from scratch *without* using any ALDP target samples. This already contrasts significantly with prior works [1,3], which assume that near-target samples are available.
> > >
> > > - As for SPICE and GEOM-DRUG, which involve larger molecules also on Cartesian coordinates, our amortized ASBS sampler does *not* use training data either. Specifically, Table 4 second last and third last rows indicate that, without any data, ASBS trained from scratch is able to surpass the performance of AS pretrained using RDKit samples. That is, by exploring a richer model class of non-memoryless processes, ASBS has successfully scaled up amortized conformer samplers without the use of any data. We intended to highlight it in L288-290:
> > >
> > >   > This highlights the significance of domain-specific priors, aiding exploration as effectively as pretraining with additional (near-target) data, which may not always be available
> > >
> > >   which we will update to make it more clear that we are explicitly showcasing ASBS’s ability to learn from only an energy model and no training data. Finally, we note that the energy model [5] for the amortized conformer benchmark was trained on SPICE, which does *not* explicitly include exhaustive conformer families and are largely off-equilibrium structures. That is, no near-target conformer data was used in the entire pipeline including energy model and training either.
> > >
> > >
> > > - We kindly note that the term “pretrain” in Sec 6 and Table 4 refers to warming up the control $u$ using RDKit samples (without relaxation), rather than the ground-truth target conformers (from CREST [6]). These RDKit samples have only 34-35% recall coverage, as shown in the first row of Table 4, thereby should *not* be considered as target samples. The term “pretrain” was adopted from the original AS paper to maintain consistency. However, we acknowledge that this term may cause confusion with “pretraining with training data”, which is *not* what we intended. In the revision, we will replace all “pretrain” with “warmup”, which more accurately reflects the procedure actually implemented. We appreciate the reviewer’s careful reading and comments, which help us further improve clarity of our paper.
> > >
> > > ---
> > >
> > > [5] Learning smooth and expressive interatomic potentials for physical property prediction
> > > [6] CREST—A program for the exploration of low-energy molecular chemical space

---

### Official Review · Reviewer_5V2q · 2025-07-03

**Clarity:** 3
**Significance:** 3
**Originality:** 3
**Rating:** 5
**Confidence:** 4

**Summary:**

This work proposes a novel method for sampling from unnormalized densities. The method extends the adjoint sampler to allow for using arbitrary source distributions while retaining the scalability of matching-based objectives. The authors formalize their method using stochastic optimal control (SOC). In particular, they propose a SOC formulation of the Schroedinger Bridge problem which is key to their new methodology. However, this formulation, or more specifically, the terminal cost of the SOC problem depends on the Schroedinger potential which is not available. As such, the authors propose an optimization scheme that alternates between learning the control via SOC and updating the Schroedinger potential. They show that this scheme is an instance of the IPF algortithm and therefore converges to the Schroedinger Bridge between source and target density. They validate the effectiveness of their method on several synthetic energy functions, Alanine dipeptide, and amortized conformer generation.

**Questions:**

- Why are LV-PIS and LV-DIS reliant on importance weights as stated in Table 1?
- To the best of my knowledge, IPF only converges to the Schroedinger Bridge for $u_0 = 0.$ How valid is the pre-taining scheme introduced in Appendix C then?
- Can the log normalizer be computed? It seems like the importance weights in (78) are intractable
- Why did the authors use such small buffer sizes?
- What symmetries did the authors use for ALDP? I could not find anything mentioned in the appendix. Also, why use internal coordinates and not Cartesian as done for the conformer generation? I reckon the harmonic prior would also help for the ALDP task
- The authors write that they use Gaussian priors for all other targets. Did they also use the corrector matching algorithm in this case? This would not be necessary since there is also a SOC formulation for Gaussian priors using DDS. A comparison would be very interesting here.
- In the Appendix it says ‘For SCLD, we do not use subtrajectory splitting’. So, this would be CMCD with LV loss then?
- How much did the harmonic prior help for the synthetic energy functions?

**Ethical Concerns:**

["NO or VERY MINOR ethics concerns only"]

**Final Justification:**

The authors have addressed all my concerns.

**Limitations:**

yes

**Quality:**

3

**Strengths And Weaknesses:**

### **Strengths And Weaknesses**

Strenghts

- Elegant SOC formulation of the Schroedinger Bridge problem
- Both SOC and corrector objectives are highly scalable
- The method is very sample efficient, which is crucial for most real-world applications

Weakesses

- Performance on ALDP (Ramachandran plot) is not very convincing, especially since internal coordinates were used. There exist several works that obtain significantly better performance, see e.g. [1].
- The paper is heavily based on adjoint sampling

[1] Temperature-Annealed Boltzmann Generators

---

> ### Author Rebuttal · Authors · 2025-07-31
>
> We thank the reviewer for raising all valuable comments and questions. We are excited that the reviewer identified the novelty of technical contributions in connecting SB with SOC, appreciated the scalability of our SB-based diffusion samplers, and the efficiency evidenced by empirical results on real-world sampling problems. We believe ASBS takes a significant step toward scalable diffusion samplers with flexible model design. We try our best to resolve all raised concerns in the responses below.
>
> ---
>
> **1. Additional experiments on ALDP**
>
> - We first thank the reviewer for raising the comments. The experiments for ALDP (Table 3 & Fig 5) were conducted in internal coordinates mainly to maintain consistency with prior SOC-based diffusion samplers, specifically PIS [2], to which our ASBS sampler also belongs. That being said, our ASBS can indeed be run on the full Cartesian coordinates for ALDP.
>
>
> - In the table below, we report preliminary results for ASBS (using harmonic source) trained from scratch in full Cartesian coordinates, without relying on any explicit target or MCMC samples. Similar to Table 3, we report the KL divergences for the 1D marginal across five torsion angles, $(\phi, \psi, \gamma_1, \gamma_2, \gamma_3)$, as well as the Wasserstein-2 on joint distribution of $(\phi, \psi)$. The latter serves as a quantitative measure of the Ramachandran plot. We kindly note that, in accordance with this year’s NeurIPS policy, we are unable to provide any external link to images or updating the manuscript.
>
>
>   |  | $\phi$ | $\psi$ | $\gamma_1$ | $\gamma_2$ | $\gamma_3$ | $(\phi,\psi)$ |
>   |---|---|---|---|---|---|---|
>   | ASBS (Cartesian) | 0.78 | 0.54 | 0.20 | 0.11 | 0.11 | 1.30 |
>
>
> - Our preliminary results imply that ASBS can generate samples with reasonable structures, with Ramachandran plots that match those of prior SOC-based diffusion samplers in internal coordinates (see Table 3). Notably, ASBS achieves this without using any explicit target samples during training, relying solely on the energy function of ALDP. This contrasts with previous attempts on ALDP in Cartesian coordinates—such as [1,3] mentioned by the reviewer and Reviewer jVAy—which often require explicit samples from or near the target distribution. We emphasize this distinction as a key advantage of adjoint-based diffusion samplers, which we further detail in the second section.
>
>
> - Finally, while we believe that the current results could be further improved with additional tuning and/or pretraining, we also highlight an intriguing application of ASBS: using model samples from [1,3] as its source distribution. This effectively transforms ASBS to a “fine-tuning” method, offering an alternative to importance sampling at inference time, which can be costly in terms of energy evaluation. We highlight this flexibility in choosing source distribution, a feature that is otherwise infeasible in prior adjoint-based methods, particularly Adjoint Sampling.
>
> ---
>
> **2. Discussion on Adjoint Sampling (AS)**
>
> - While our ASBS is closely related to AS, as noted by the reviewer, we emphasize that the motivation behind ASBS differs significantly from that of AS. Practically, AS can only be used for sampling from a distribution, while ASBS can be used to transport from any (e.g. a similar-but-incorrect) distribution. Despite this additional capability, we found ASBS also outperforms AS in pure sampling due to the less restrictive requirements. Theoretically, ASBS is designed to solve the Schrödinger Bridge (SB) problem (Eq 3) whereas AS solves a specific form of SOC problem (Eq 7). The theoretical connection between these two problems only became clear with the presentation of Thm 3.1. This is also acknowledged by the reviewer, i.e., "Elegant SOC formulation of SB problem”.
>
>
> - Given the practical success of AS in terms of both performance and energy efficiency, along with our theoretical results in Thm 3.1 that generalizes AS to a broader, non-memoryless SOC problem, it is natural to pursue a new learning method that retains the algorithmic advantage of AS while addressing additional complexity from non-memorylessness and arbitrary source distributions. As shown in Sec 3.3, this can be achieved by incorporating an additional corrector matching objective (Eq 15). Since Eq 15 employs a simple matching objective, the overall ASBS method (Alg 1) remains as scalable as AS (Fig 4).
>
>
> - Such a straightforward algorithmic modification from AS to ASBS, although theoretically complex, has led to significant performance gains, evidenced by all of our experiments (Tables 2,3,4). Furthermore, it does not impede convergence to global solution (Thm 3.2). To summarize, we believe ASBS complements AS with additional flexibility in selecting informative sources and utilizing richer model classes, leading to significant performance gains without compromising algorithmic scalability. Together, these advancements enable ASBS to achieve strong performance and open new avenues for future applications.
>
>
> ---
>
>
> **3. Clarifications on questions**
>
> - **LV-PIS & LV-DDS in Table 1.** We thank the reviewer for raising this question. Our intention was to convey that LV directly minimizes the variance of the importance weights (IWs), and thus, methods in this category involve the computation of IWs. We acknowledge that they do not perform importance sampling, which may have been incorrectly inferred from the current Table 1. We will clarify this in Table 1 in the revision.
>
>
> - **Pretraining with IPF.** As the reviewer conjectured, the global convergence of IPF requires initialization from either $u=0$ or $h=0$. Initializing IPF with a pretrained $u$—which corresponds to the first stage of Iterative Markovian Fitting (IMF)—has been theoretically explored recently [4] and shown to maintain global convergence in Gaussian setups. However, in more complex scenarios, we still observe stable local convergence. Alternatively, we can initialize ASBS by constructing its source distribution using samples from pretrained models, thereby leveraging ASBS's flexibility in selecting source distributions without compromising global convergence.
>
>
> - **Log normalizer of ASBS.** The importance weights in Eq 78 can be computed by re-parametrizing $u$ and $h$ with two potential networks, specifically $u = \nabla v$ and $h = \nabla \bar h$ as shown in Eq 80, and then calculating the importance weights using Eqs 78 & 81. Alternatively, we can train an additional backward drift $v_t := \nabla \log \hat \varphi_t$ with standard SB matching [5], and then use Eq 79 to compute the ratio $\log \frac{\hat \varphi_1}{\hat \varphi_0}$. While both methods are tractable, they introduce additional complexity. Exploring alternative approaches to compute the weights would be a promising direction for future research (L295).
>
>
> - **Buffer size in Table 5.** This’s a typo. The correct buffer size $|\mathcal{B}|$ for conformer generation should be 64000, which is consistent with the order of magnitude used for other tasks. We thank the reviewer for the meticulous reading!
>
>
> - **Symmetry for ALDP.** In practice, we utilize the full internal coordinate transform from FAB [6], which converts Cartesian coordinates into bond lengths, bond angles, and dihedral angles, effectively removing 6 degrees of freedom (3 for translation and another 3 for rotation). Non-augular coordinates are further normalized using samples with minimal energies. We refer the reviewer to Appendix F.1 in [6] for further details and will include these discussions in the revised Appendix.
>
>
> - **Corrector matching (CM) for Gaussian prior.** We first clarify that CM is used for all ASBS models, including those with Gaussian priors. This is because the base processes $p^\text{base}$ for all ASBS models are not memoryless—with $f :=0$ and $\sigma_t$ set according to Table 5— hence necessitating the use of CM, as discussed in Sec 3.2. As the reviewer conjectured, there is indeed a memoryless base process (specifically, the one considered in DDS) for Gaussian prior, which eliminates the need for CM. This corresponds exactly to the leftmost plot in Fig 1, where significant noise is injected into sampling to ensure the conditional independence of the memoryless process (Eq 6). We have also reported its performance (i.e., DDS) on synthetic energies (Table 2) and ALDP (Table 3), where ASBS outperformed DDS in both cases. We will include these clarifications in the revision.
>
>
> - **SCLD.** We thank the reviewer for catching the error. We did use subtrajectory splitting (with the default value of 4) for SCLD, so it does not degenerate to CMCD with LV. In practice, we find that using subtrajectories yields better results. This will be corrected in the revision, and we appreciate the reviewer’s meticulous reading.
>
>
> - **Harmonic vs Gaussian prior in synthetic energies.** In the below table, we present a comparison of ASBS using harmonic and Gaussian priors. The results are consistent with those in Table 4, showing slight improvements with harmonic priors in both cases.
>
>   |  | DW4 (W2) | DW4 (EW2) | LJ13 (W2) | LJ13 (EW2) | LJ55 (W2) | LJ55 (EW2) |
>   |---|---|---|---|---|---|---|
>   | Harmonic | 0.38 | 0.19 | 1.59 | 1.28 | 4.00 | 27.69 |
>   | Gaussian | 0.42 | 0.20 | 1.60 | 1.82 | 4.09 | 38.70 |
>
>
> ---
>
> [1] Temperature-Annealed BG
> [2] Path Integral Sampler
> [3] Scalable Equilibrium Sampling with Sequential BG
> [4] Diffusion & Adversarial SB via IPMF
> [5] Diffusion SB Matching
> [6] Flow Annealed Importance Sampling Bootstrap

---

> > ### Comment · Reviewer_5V2q · 2025-08-04
> > **Acknowledgement**
> >
> > We thank the authors for their detailed reply, which has addressed the concerns. We increased the rating accordingly.

---

### Official Review · Reviewer_RKyV · 2025-07-06

**Clarity:** 3
**Significance:** 3
**Originality:** 3
**Rating:** 5
**Confidence:** 3

**Summary:**

The authors introduced the Adjoint Schrödinger Bridge Sampler (ASBS), which is a novel diffusion-based sampler for Boltzmann distributions that addresses general Schrödinger bridge sampling problems using only the target energy function.

**Questions:**

See comments on Weaknesses.

**Ethical Concerns:**

["NO or VERY MINOR ethics concerns only"]

**Final Justification:**

I thank the authors for their detailed and thoughtful rebuttal. They have convincingly addressed all of my initial concerns, and I appreciate the detailed clarifications provided. Based on this strong response and the authors' clear plan for revision, I have raised my score to 5.

**Limitations:**

Yes

**Paper Formatting Concerns:**

No major formatting concerns.

**Quality:**

3

**Strengths And Weaknesses:**

Strengths:

It is known that the SOC setup for fine-tuning is limited to the Dirac measure [1] [2] for the initial distribution.
The authors extended it to handle arbitrary distributions with memoryless source distributions by leveraging Schrodinger bridge between two marginal distributions. Schrodinger bridge (SB) should be the right tool here.

Extensive evaluations are conducted with promising performance.

[1] Theoretical guarantees for sampling and inference in generative models with latent diffusions

[2] Path Integral Sampler: a stochastic control approach for sampling

Weaknesses:

I understand that SB are complex to understand. Howevere, the writings are a bit sloppy and many important details are less evalborated, e.g.
	1) Eq.(5) is quite important to understand why we should avoid memory setups and the authors at least should cite the adjoint matching paper. To be honest, even for the adjoint matching paper, some details are not detailed or derived properly as well, which makes it harder to understand.
	2) eq(6), if p^base is a distribution that takes a joint distribution, then we should not reuse it a non-joint distribution; the intuition of setting g(x):=log p_1^base / nu(x) is not elaborated.
	3) eq(14) eq(15) are not evalaborated clearly and it is crucial to understand the dynamics and why it works, please write it in more details.

Didn't check proof.

---

> ### Author Rebuttal · Authors · 2025-07-31
>
> We thank the reviewer for raising all valuable comments. We are excited that the reviewer identified the novelty of our technical contributions for overcoming Dirac limitations from prior methods, appreciated the advantages of SB-based diffusion samplers, and acknowledged our strong and extensive empirical results. We believe ASBS takes a significant step toward scalable diffusion samplers with flexible model design. We try our best to resolve all raised concerns in the responses below.
>
> ---
>
> **1. Clarification on Sec 2 (Eqs 5 & 6)**
>
> - We understand that current Sec 2 may be compactly written, mainly due to the space constraints at submission. In response to the reviewer’s suggestion, we will provide further clarifications in the revised Sec 2, along with a self-contained, step-by-step derivation in the revised Appendix. We detail these clarification below.
>
>
> - The goal of Sec 2, as briefly stated in L86-88, is to formally introduce the mathematical formulation of the SOC problem (Eq 4), its optimal distribution (Eq 5), and how prior diffusion samplers have adopted it for Boltzmann sampling under the memoryless condition (Eq 6). This leads to a specialized, “_memoryless_”, SOC problem (Eq 7), which contrasts with our generalized, “_non-memoryless_”, SOC problem inspired by Schrödinger Bridge (Eq 9; also see L129-131).
>
>
> - As the reviewer noticed, Eq 5 is indeed the key to connect SOC problems with Boltzmann sampling. This equation, representing the optimal solution to the original SOC problem (Eq 4), can be derived as follows:
>   - By rewriting Eq 4 as KL-regularized optimization, $\min_{p^u} \mathrm{KL}(p^u || p^\text{base}) + \mathbb{E}_{p^u_1}[g(X_1)]$ using Girsanov theorem [1].
>   - Due to the KL term, the aforementioned optimization is convex in $p^u$, allowing for an analytic solution. Setting the first-order variation to zero yields $p^\star(X_1 | X_0) \propto p^\text{base}(X_1 | X_0) e^{-g(X_1)}$.
>   - Finally, substituting the normalization constant $\int p^\text{base}(X_1 | X_0) e^{-g(X_1)} \mathrm{d} X_1 = e^{-V_0(X_0)}$ leads to Eq 5.
>
>   While Eq 5 is well-known in SOC literature [2,3], we acknowledge that it may not be immediately clear to readers without sufficient background—even with the interpretation provided right below (L93-96). In the revision, we will include these clarifications along with additional citations for Eq 5, including the original AM paper, which—we kindly note that—was already cited in the beginning of Sec 2 (L86).
>
>
> - Eq 5 implies that, for any SOC problems (Eq 4) to be effective in sampling Boltzmann distribution, the optimal distribution $p^\star$ must satisfy the desired target distribution $\nu$ at terminal time $t=1$. This requires appropriately selecting the base drift $f_t$, noise schedule $\sigma_t$, source distribution $\mu$, and terminal cost $g$, such that $\int p^\star(X_0, X_1) \mathrm{d} X_0 = \nu(X_1)$. In Adjoint Sampling [4], this is achieved by
>   - Enforcing the base process $p^\text{base}$—uniquely defined by $(f_t, \sigma_t, \mu)$—to be memoryless (Eq 6). This simplifies the marginalization to $\int p^\star(X_0, X_1) \mathrm{d} X_0 = C \cdot p^\text{base}(X_1)_1 e^{-g(X_1)}$ with some constant $C$, as detailed between L102-123.
>   - Finally, equating $p^\text{base}_1(X_1) e^{-g(X_1)} = \nu(X_1)$, which leads to the proposed terminal cost $g := \log p^\text{base}_1 - \log \nu$ in L102.
>
>   We will include these clarifications for how $g$ is set up in the revision, and we thank the reviewer for pointing it out.
>
>
> - Finally, for notational simplicity, we omitted the time specification of distribution—such as $p^\text{base}(X_0)$ and $p^\text{base}(X_1)$ in Eq 6—whenever it can be clearly inferred from the variables $X_0$ and $X_1$. As the reviewer correctly pointed, this is not mathematically rigorous, and we will correct these inconsistencies in the revision.
>
> ---
>
> **2. Clarification on the Adjoint Matching (Eq 14) and Corrector Matching (Eq 15)**
>
> - Given our proposed SOC problem (Eq 9), which accommodates non-memoryless conditions and arbitrary source distributions, Sec 3.2 identifies two key mathematical objects, namely the optimal control $u^\star$ in Eq 12 and the corrector $\nabla \log \hat \varphi_1$ in Eq 13. These two mathematical objects characterize the optimality condition of our SOC problem (Eq 9) and, equivalently, the Schrödinger Bridge problem (Eq 3). In other words, the diffusion sampler that solves Eqs 9 & 3 must satisfy the optimality conditions in Eqs 12 & 13.
>
> - Building on the reasoning above, our algorithmic goal is to design a learning method that finds a diffusion sampler satisfying Eqs 12 & 13, which correspond to two simple matching-based objectives (RHS of Eqs 12 & 13). However, these two matching objectives cannot be implemented naively due to their interdependency: Solving the matching objective in Eq 12 for the optimal control $u^\star$ requires knowledge of the corrector $\nabla \log \hat \varphi_1$. Likewise, solving the matching objective in Eq 13 for $\nabla \log \hat \varphi_1$ requires knowledge of $u^\star$.
>
>
> - In Sec 3.3, we address the interdependency between the control $u$ and the corrector $h$ through an alternating training scheme. Specifically,
>   - Starting from an initial guess $(u, h)$ of the control and corrector, we first update $u$ by solving Eq 12 with $h \approx \nabla \log \hat \varphi_1$.
>   - Then, we update $h$ by solving Eq 13 with $u \approx u^\star$.
>   - This alternate update between $u$ and $h$ continues until $u$ converges to a fixed point (see Alg 1).
>
>   This effectively creates a sequence of update, $(u^{(0)}, h^{(0)})$ → … $(u^{(k)}, h^{(k)})$ → …, that may be thought of as running coordinate descent between the control $u$ and the corrector $h$. Intuitively, at each stage $k$, we first find the control $u^{(k)}$ that best aligns with the corrector from previous stage, $h^{(k-1)}$, then update the corrector $h^{(k)}$ accordingly to reflect the “memorylessness” of the current control $u^{(k)}$. We formalize these update rules in Eqs 14 & 15, and their dynamics are visualized on a 2D toy example in Fig 2.
>
>
> - Whether the aforementioned update sequence converges—and if so, whether it converges to the global solution—is a highly non-trivial question. Nevertheless, we provide a positive answer in Thm 3.2. The proof, detailed in Sec 4, essentially shows that the update rules in Eqs 14 & 15 are contractive maps that progressively bring $u^{(k)}$ closer to the optimal control $u^\star$. Formally, the KL divergence $\mathrm{KL}(p^u_1 || \nu)$ is non-increasing as $k$ grows [5].
>
>
> Overall, we acknowledge that Sec 3.3 is compactly written and will include these discussions in the revision. We thank the reviewer again for the suggestion.
>
>
> ---
>
> [1] Applied SDEs (Särkkä & Solin, 2019)
> [2] Adjoint Matching
> [3] Solving high-dim HJB PDEs using neural networks
> [4] Adjoint Sampling
> [5] Diffusion SB with App. to Score-Based Generative Modeling

---

> ### Author Response · Authors · 2025-08-07
> **Remind of author-reviewer discussion period**
>
> We thank the reviewer again for all valuable comments. In our rebuttal, we have clarified all raised concerns by providing self-contained, step-by-step derivations of Eqs (5), (6), (14), and (15). These clarifications will be included in the revised Sec 2 and Appendix. We appreciate the reviewer’s feedback in enhancing the clarity of our presentation—a key feature that we highly value and has been positively recognized by Reviewers jVAy & NUeM.
>
> As we approach the end of the discussion period (**Aug 8th 11:59pm AOE**), we would like to ask whether our responses have covered the questions raised in your initial reviews. We are happy to provide any further clarification and discussion. If our replies adequately address your concerns, we kindly ask the reviewer to reconsider their initial rating, so that it better reflects the discussion at the current stage.

---

### Decision · Program_Chairs · 2025-09-17

**Decision:**

Accept (oral)

**Comment:**

This paper introduces the Adjoint Schrödinger Bridge Sampler (ASBS), a diffusion-based framework for sampling from unnormalized densities, built on a refined formulation of stochastic optimal control and the Schrödinger bridge. All reviewers agree that this work is exceptionally strong and merits oral presentation.

The contributions of this paper are groundbreaking. It advances the theoretical foundations of diffusion-based sampling with a bold relaxation of the memoryless condition in adjoint sampling, opening new directions for the field. The algorithmic formulation is not only elegant but also conceptually unifying, drawing connections to classical iterative proportional fitting in optimal transport while extending them in novel ways. On the practical side, the method achieves remarkable scalability and sample efficiency, setting a new benchmark for tackling complex multimodal distributions.

What makes this paper stand out is its rare combination of strengths across the full spectrum: rigorous theory, innovative algorithms, practical relevance, and convincing experiments. The empirical results are extensive and show the method's unique ability to navigate diverse modes, including those far from initialization, a clear indication of its power and robustness. The reviewers also highlighted the excellent clarity of exposition and the thoughtful rebuttal, which successfully addressed all initial concerns.

In conclusion, this is a groundbreaking submission that combines theoretical depth, algorithmic innovation, and practical effectiveness. It is likely to shape future research in this area, and I strongly recommend acceptance as an oral.